# The genomic landscape of 85 advanced neuroendocrine neoplasms reveals subtype-heterogeneity and potential therapeutic targets

Job van Riet [1,2,3,11], Harmen J. G. van de Werken [1,2,11✉], Edwin Cuppen[4,5], Ferry A. L. M. Eskens[3], Margot Tesselaar[6], Linde M. van Veenendaal[6], Heinz-Josef Klümpen[7], Marcus W. Dercksen[8], Gerlof D. Valk[9], Martijn P. Lolkema [3,10], Stefan Sleijfer[3,10] & Bianca Mostert [3✉]

Metastatic and locally-advanced neuroendocrine neoplasms (aNEN) form clinically and genetically heterogeneous malignancies, characterized by distinct prognoses based upon primary tumor localization, functionality, grade, proliferation index and diverse outcomes to treatment. Here, we report the mutational landscape of 85 whole-genome sequenced aNEN. This landscape reveals distinct genomic subpopulations of aNEN based on primary localization and differentiation grade; we observe relatively high tumor mutational burdens (TMB) in neuroendocrine carcinoma (average 5.45 somatic mutations per megabase) with TP53, KRAS, RB1, CSMD3, APC, CSMD1, LRATD2, TRRAP and MYC as major drivers versus an overall low TMB in neuroendocrine tumors (1.09). Furthermore, we observe distinct drivers which are enriched in somatic aberrations in pancreatic (MEN1, ATRX, DAXX, DMD and CREBBP) and midgut-derived neuroendocrine tumors (CDKN1B). Finally, 49% of aNEN patients reveal potential therapeutic targets based upon actionable (and responsive) somatic aberrations within their genome; potentially directing improvements in aNEN treatment strategies.

[1] Cancer Computational Biology Center, Erasmus MC Cancer Institute, University Medical Center, Rotterdam, the Netherlands. [2] Department of Urology, Erasmus MC Cancer Institute, University Medical Center, Rotterdam, the Netherlands. [3] Department of Medical Oncology, Erasmus MC Cancer Institute, Rotterdam, the Netherlands. [4] Center for Molecular Medicine and Oncode Institute, University Medical Center Utrecht, Utrecht, the Netherlands. [5] Hartwig Medical Foundation, Amsterdam, the Netherlands. [6] Department of Medical Oncology, Cancer Institute, University of Amsterdam, Amsterdam, The Netherlands. [7] Department of Medical Oncology, Amsterdam University Medical Centers, Cancer Center Amsterdam, Amsterdam, The Netherlands. [8] Department of Internal Medicine, Maxima Medisch Centrum, Veldhoven, The Netherlands. [9] Department of Endocrine Oncology, University Medical Center Utrecht, Utrecht, The Netherlands. [10] Center for Personalized Cancer Treatment, Rotterdam, the Netherlands. [11] These authors contributed equally: Job van Riet, Harmen J. G. van de Werken. ✉email: h.vandewerken@erasmusmc.nl; b.mostert@erasmusmc.nl

Neuroendocrine neoplasms (NEN) are a heterogeneous and uncommon tumor type. It can arise from any of the neuroendocrine cells distributed widely throughout the body. As outlaid by the International Agency for Research on Cancer and World Health Organization, a clinical distinction is made between the poorly differentiated neuroendocrine carcinomas (NEC) and the more differentiated neuroendocrine tumors (NET)[1,2], the latter are further subdivided based on their primary site in pancreas (pNET), gastro-intestinal tract or lung. Further distinctions are made based upon grade (as assessed by Ki-67 or MIB-1 staining as a measure of proliferation index), differentiation, histology (small-cell vs. large-cell) and functionality (the presence or absence of hormone secretion resulting in typical clinical syndromes dependent upon the predominant hormone that is secreted). Tumor grade and differentiation are associated with prognosis, and all the aforementioned factors affect the choice of treatment. However, also in small subgroups of NEN, such as well-differentiated low-proliferating pNET, marked clinical and genetic heterogeneity occur, as well as vastly different responses to treatment with only few mutant genes such as *DAXX*, *ATRX*, and *MEN1* serving as prognostic markers[3–6]. Thus, the parameters by which NEN are currently classified do not sufficiently separate patients and tumors according to prognosis and response to therapy. Nonetheless, certain anti-tumor therapies (i.e., sunitinib and everolimus) have been registered for distinct NEN-subtypes. Hence, there is a high unmet need to better classify and understand these diverse tumors, ultimately leading to more tumor- or patient-tailored therapeutic strategies.

Thus far, limited whole-exome sequencing (WES) and whole-genome sequencing (WGS) data are available for NEN, probably reflecting the rarity of this disease. Currently, pNET have been characterized most extensively; 81 primary tumors were subjected to WGS as part of the PCAWG project[7] and another set of primary pNET (*n* = 102) was described by Scarpa et al.[5]. In addition, smaller series using diverse sequencing approaches of varying resolution on primary NET subtypes have been published; which include genomic studies on pNET (WES and targeted sequencing; *n* = 10 and 58, respectively)[4], DNA methylation and RNA-sequencing of pNET (*n* = 32 and *n* = 33, respectively)[3], well-differentiated carcinoid (SNP-array; *n* = 29)[8], NEC (targeted sequencing; *n* = 63)[9] and two studies on a multi-institution cohort of small intestine NET (SI-NET) using combined approaches of targeted sequencing (*n* = 81), WES (*n* = 48; *n* = 29) and WGS (*n* = 15)[10,11]. These studies have shown that NET have a relatively stable genome and only few commonly observed driver mutations and allelic imbalances, often associated with their primary tissue of origin. Previously associated genetic drivers of NET include the cell-cycle regulator *CDKN1B* in SI-NET[10–13], chromatin-remodeling genes (*DAXX*, *ATRX*, *MEN1*, and *SETD2)*, DNA-repair genes (*CHEK2*, *BRCA2*, and *MUTYH*), mTOR-related genes (*TSC2*, *PTEN*, and *PIK3CA*) and the oxygen-sensing modulator *VHL*[14] in pNET[3–5,7,15] whilst NEC is associated with aberrations in *TP53*, *RB1*, *MYC*, *CCND1*, *KRAS*, *PIK3CA/PTEN* and *BRAF*[9,16–18]. However, these studies were all performed on primary tumor specimens, whilst a patient generally dies from the consequences of metastatic disease. In addition, we know from other tumor types that marked heterogeneity can occur between primary and metastatic tumor cells[19–22], due to inherent genomic instability and/or the influence of targeted or cytotoxic treatment on the tumor genome. These discrepancies should be taken into account when assessing a patient's prognosis and possible treatment options, and can be better understood through thorough genomic characterization of metastases. To date, analysis of metastatic NET is limited to two studies describing series of five patients with NET originating in the pancreas and the small intestine (or midgut), respectively[23,24].

These studies have shown focal amplification of *MYCN* concomitant with loss of *APC* and *TP53* in one sample as important metastatic genetic aberrations. For NECs, only two series of WGS of the primary tumors of (1) five cervical and (2) 12 genitourinary NECs have been published[25,26].

In this work, WGS was performed on 85 biopsies from patients with locally advanced or metastatic (advanced) NEN (aNEN); a single biopsy per patient was selected for analysis. The vast majority of these biopsies are taken from metastatic lesions (*n* = 70 out of 85) whilst for 15 patients suffering from metastatic or incurable locally advanced disease, their treating physician judged a biopsy of a metastasis too high-risk or not feasible, and instead had a biopsy taken from their primary lesion at the time of locally advanced or metastatic disease. All aNEN patients underwent these biopsies as part of their participation in the Dutch CPCT-02 and DRUP studies[27,28]. We report on the presence of genomic alterations, mutational and rearrangement signatures for the whole aNEN cohort and reveal genomic characteristics and alterations distinguishing aNEC from aNET. Furthermore, we make a genomic distinction between pancreas- and midgut-derived aNET. In addition, we investigate the presence of actionable genetic alteration within aNEN patients, which might render them eligible for off-label or experimental systemic treatments to extend therapy options.

## Results

**Overview of included patients within the CPCT-02 aNEN cohort and whole-genome sequencing.** A total of 108 patients, originally classified as having a neuroendocrine neoplasm, were included in the CPCT-02 and DRUP studies and had a primary or metastatic tumor biopsy taken in parallel with a blood control (Fig. 1). Five patients were excluded because of missing or withdrawn informed consent, and another five had non-evaluable biopsies due to low (<20%) tumor cell percentage or low DNA yield. Thirteen biopsies were excluded because of incomplete clinical records, misclassifications of the tumor (based on additional checks of the medical records), or were duplicate biopsies from the same patient. An overview of the aNEN patient inclusion per participating Dutch center (*n* = 13) can be found in Supplementary Fig. 1a.

The aNEN cohort is represented by 37 females and 48 males with a median age of 62 (Q(uartile)$_1$–Q$_3$: 57–68) and 61 (Q$_1$–Q$_3$: 56–68) years, at time of biopsy, respectively (Fig. 1c). In total, 69 NET and 16 NEC were included. The primary tumor location in the midgut was most common (*n* = 41, 48%), followed by pancreas (*n* = 23, 27%) and unknown (*n* = 12, 14%) (Fig. 1b). Most of the tumor biopsies were taken from liver metastases, and a minority from relapses at the primary site (Fig. 1d).

To gain more in-depth knowledge of the pathological information of this cohort, we requested pathological reports of primary tumor and/or metastatic tumor tissue as available in the nationwide (Dutch) PALGA registry. Of note, these tissues were often not acquired at the time of biopsy for the CPCT-02 study. For the majority of patients, pathology reports on metastases and/or the primary tumor were available (Supplementary Data 1). In the minority, the pathological record of a previous primary biopsy or resection specimen was assessed.

We also characterized our cohort with regard to previously administered systemic anti-tumor treatment. Sixty-nine percent of patients had not undergone any previous anti-tumor treatment, 31% had undergone a large variety of previous treatments, mainly consisting of somatostatin analogs, radioisotopes, chemotherapy, and targeted therapy (Supplementary Fig. 1d).

The tumor biopsies and corresponding peripheral blood controls from the 85 distinct patients were whole-genome sequenced using paired-end protocols, to a median mean read

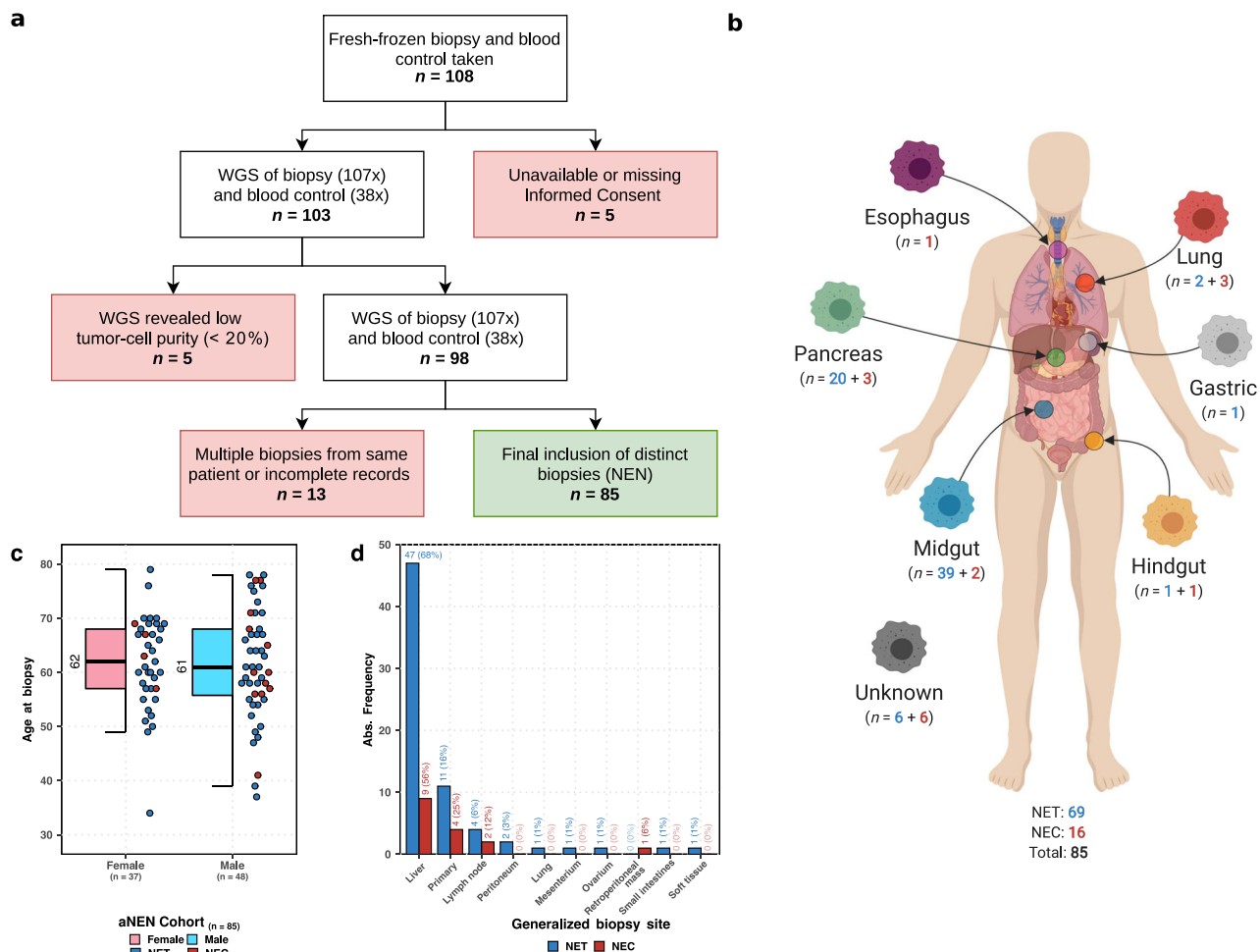

**Fig. 1 Overview of patient inclusion and subclassification of biopsies. a** Flowchart of patient inclusion. From the CPCT-02 cohort, single biopsies from 85 distinct patients with advanced (metastatic or locally advanced) neuroendocrine neoplasms (aNEN) were selected. From the total pool of available whole-genome sequenced aNET samples. If multiple derived aNET biopsies from the same patient were available, we selected the aNET biopsy with the highest tumor cell purity. The tumor and matching blood sample (reference) were whole-genome sequenced to a median read coverage of 107 and 38 (paired-end) reads per base, respectively. Filtering criteria in which patients were excluded are highlighted in red. The final inclusion of aNEN patients is depicted in green. **b** Subclassification of aNEN based on primary localization. The 85 aNEN were subclassified, based on their primary localization, into six major categories; gastric, hindgut, lung, esophagus, pancreas, and midgut; whilst samples with indeterminable localization were categorized as unknown. The number of aNET (in blue) and aNEC (in red) are shown per category. **c** Age distribution stratified by gender of the aNEN cohort. Observed median per variable displayed in a boxplot with individual data points (aNET and aNEC are depicted as blue and red points respectively). The median, interquartile range (IQR), and 1.5× the IQR are represented by a solid black line, box, and whiskers, respectively. **d** Barplot of generalized location of the tumor biopsy. Absolute and relative (in brackets) frequency of aNET (blue) and aNEC (red) biopsy locations. Credit: Created with BioRender (https://biorender.com/).

coverage of 107× (Q(uartile)$_1$–Q$_3$: 99×–116×) and 38x (Q$_1$–Q$_3$: 35–42×), respectively to a median in silico estimated tumor cell purity of 0.7 (Q$_1$–Q$_3$: 0.5–0.82).

**The mutational landscape of advanced neuroendocrine neoplasms reveals differences related to primary localization and degree of differentiation.** The overall mutational landscape of aNEN ($n = 85$; Fig. 2) reveals two strikingly distinct genomic populations of neuroendocrine neoplasms, i.e., the aNEC and aNET populations. The aNEC ($n = 16$) reveals diploid to triploid genomes and a median tumor mutational burden (TMB) of 5.45 somatic mutations per Mb (Q$_1$–Q$_3$: 3.84–8.85), which is in the mid-range of TMB known for human primary cancers[29]. However, the aNET ($n = 69$) are hallmarked by a relatively stable diploid tumor genome with only few, but specific, chromosomal arm aberrations and harbors the lowest overall TMB of only 1.09 (Q$_1$–Q$_3$: 0.79–1.52) of all metastatic cohorts within the CPCT-02 study[27].

The somatically acquired and whole-genomic mutational landscape of aNEC ($n = 16$) revealed a median of 13,996 single-nucleotide variants (SNVs; Q$_1$–Q$_3$: 9465–22,830), 1.756 small insertions and deletions (InDels; Q$_1$–Q$_3$: 752–2,245), 114 multiple-nucleotide variants (MNVs; Q$_1$–Q$_3$: 49–198), 150 structural variants (SVs; Q1–Q3: 82–264) and an overall diploid to triploid genome (Q$_1$–Q$_3$: 1.9–3.1; Supplementary Fig. 2). Concordant with the lower TMB of the aNET ($n = 69$), the aNET revealed a median of 2870 SNVs (Q$_1$–Q$_3$: 1995–3904), 254 InDels (Q$_1$–Q$_3$: 185–325), 19 MNVs (Q$_1$–Q$_3$: 12–27), 17 SVs (Q$_1$–Q$_3$: 7–53) and an overall diploid genome (Q$_1$–Q$_3$: 1.9–2.19). The discrepancy in mutational load between aNEC and aNET also held true when inspecting only the coding regions, in which aNEC revealed a higher number of SNVs, InDels, MNV compared to aNET (Supplementary Fig. 2a). Similarly, aNEC displayed elevated numbers of all SV classes (translocations, deletions, tandem duplications, insertions and inversions; Supplementary Fig. 2d).

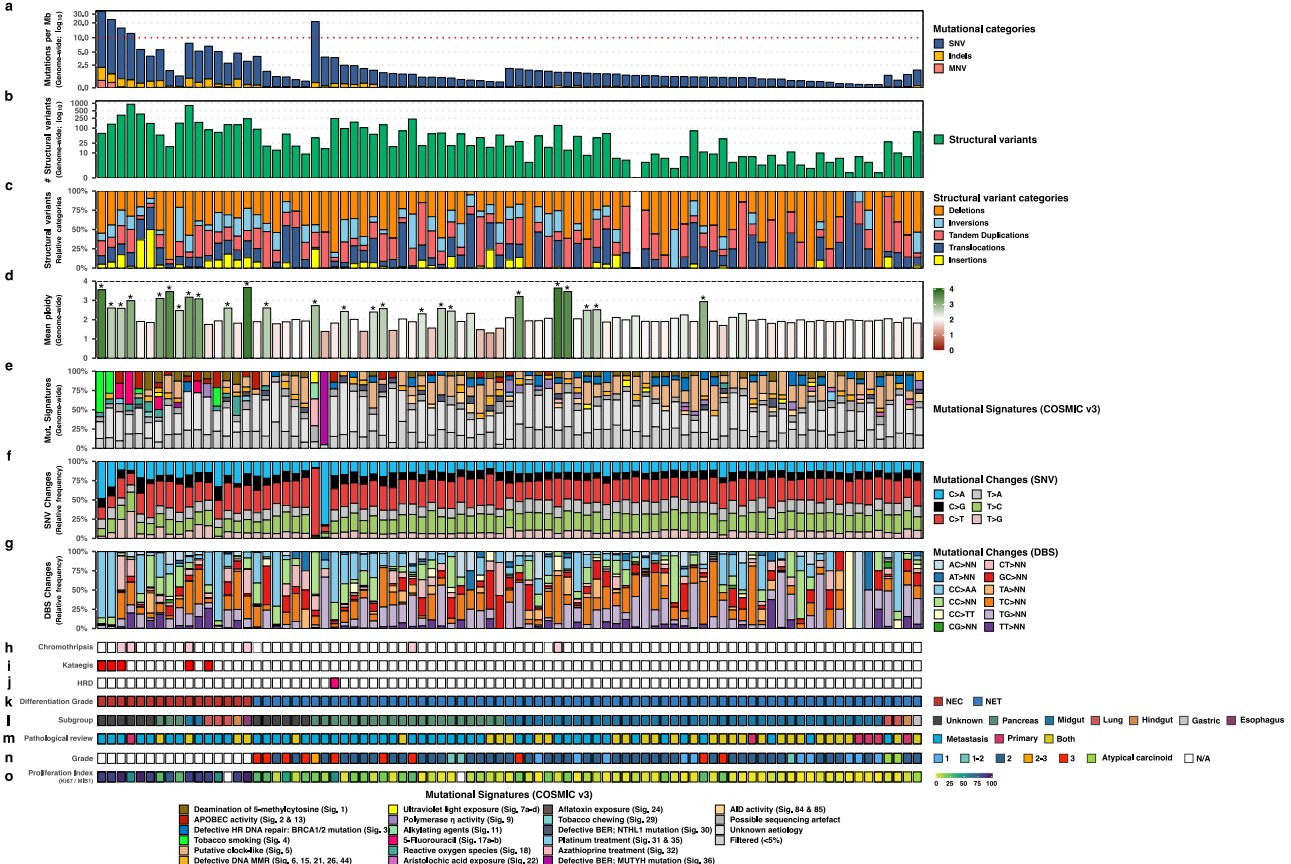

**Fig. 2 Landscape of large-scale genomic alterations detected in aNEN, ordered by differentiation grade (NEC/NET) and primary localization.** Overview of genome-wide characteristics of the aNEN cohort ordered by aNEC/aNET and primary localization on decreasing median tumor mutational burden. For each aNEN ($n = 85$), the following tracks are shown: **a** Number of genomic mutations per megabase over the entire genome (TMB). Threshold for high TMB ($\geq 10$) is shown by a horizontal red dotted line. Y-axis is shown in $\log_{10}$-scale. **b** Total number of structural variants (green) including deletions, tandem duplications, translocations, inversions, and insertions as detected by GRIDSS. Y-axis is shown in $\log_{10}$-scale. **c** Relative frequency of each of the structural variant categories; deletions in orange, tandem duplications in red, translocations in blue, inversions in light-blue, and insertions in yellow. **d** Mean genome-wide ploidy, ranging from 0 (red) to 4 (green; tetraploid). Common diploid status is shown in white. Suspected whole-genome duplication (WGD) events have been marked by an asterisk (*). **e** Relative contribution of the COSMIC single-base substitution mutational signatures (v3; $n = 67$). Proposed etiology of signatures is denoted below. **f** Relative frequency of the pyrimidine point mutations (SNV) in their six categories. **g** Relative frequency of Doublet Base Substitution (DBS) categories. **h** Presence of chromothripsis; aNEN with chromothripsis are shown in pink. **i** Presence of kataegis; aNEN with $\geq 1$ kataegis events are shown in red. **j** Status of homologous recombination deficiency (HRD), as determined by CHORD; aNEN with *BRCA1/2*-associated HRD are shown in red. **k** Differentiation grade of the aNEN; aNEC in red, aNET in blue. **l** Primary localization of the aNEN. **m** Origin of the respective pathological record used to determine differentiation grade and proliferation index. **n** Differentiation grade based on the pathological record: 1 (sky-blue), 1–2 (teal), 2 (dark blue), 2–3 (orange), 3 (red), atypical carcinoid (green) and not available (N/A; white). **o** Proliferation index (KI67 / MIB1) from 0 to 100 based on the pathological record.

The majority of somatic coding mutations for all aNEC and all aNET ($n = 3333$ and $3663$; SNV, InDel, and MNV) were found to be predicted missense variants (52% in aNEC vs. 52% in aNET), followed by synonymous variants (18% vs. 21%). The number of genes harboring somatic mutations within their coding regions differed between aNEC and aNET. Over the entire aNEC cohort ($n = 16$), 2845 distinct mutant genes were observed, versus 3112 distinct genes within the entire aNET cohort ($n = 69$). Per sample, a median of 150 ($Q_1$–$Q_3$: 127–270) versus 37 (median; $Q_1$-$Q_3$: 26–51) genes harboring mutations within coding regions were observed for aNEC and aNET samples, respectively; revealing that aNEC harbor greater numbers of mutant genes compared to aNET.

The median genome-wide ratio of transitions (Ti; A ↔ G or T ↔ C) to transversions (Tv; C ↔ A, C ↔ G, T ↔ A or T ↔ G) within aNEC was found to be 0.78 Ti\Tv ($Q_1$–$Q_3$: 0.72–1.02) vs.

1.52 Ti\Tv ($Q_1$–$Q_3$: 1.12–2.20) in the coding regions. For aNET the median genome-wide and coding Ti\Tv were found to be 1.09 ($Q_1$-$Q_3$: 0.98–1.32) and 1.42 ($Q_1$-$Q_3$: 1–1.96), respectively (Supplementary Fig. 2f).

High-TMB ($\geq 10$) are often associated with DNA-repair deficiency and/or tumors with sensitivity for immune therapy, e.g., checkpoint inhibitors. Four aNEC samples, all from unknown origin, and a single pancreatic aNET showed this high-TMB genotype (Fig. 2a). One aNET displayed signs of *BRCA2*-associated homologous recombination deficiency (HRD), as determined using the CHORD classifier which is mainly based on deletions with flanking microhomology and 1–100 kb structural duplications (Fig. 2j; Supplementary Fig. 3). Further inspection revealed that this aNET harbored a somatic frameshift mutation within *RAD51C*, a known HRD-associated gene[30–33].

**Regional hypermutation (kataegis).** Regional hypermutation (kataegis) was detected in five aNEC; Fig. 2i; Supplementary Fig. 4). Canonically, kataegis is associated with APOBEC activity and indeed, four out of five (80%) of these kataegis events predominantly showed the canonical TpCpW context associated with *APOBEC* alterations[34]. In addition, in the five samples harboring kataegis, the absolute contribution of APOBEC single-base substitution (SBS) mutational signatures (2 & 13) was significantly higher (median 45 vs. 533; $p < 0.01$; Wilcoxon rank-sum test) compared to aNEN without kataegis ($n = 80$).

**Chromothripsis.** Multiple distinct aNEN (four aNEC and two aNET; 7%) revealed the presence of chromothripsis, a catastrophic phenomenon of the shattering and interchromosomal recombination of one or more chromosomes (Fig. 2h; Supplementary Fig. 5). Strikingly, four of the six observed chromothripsis events from distinct aNEN (two aNEC and two aNET) involved the same chromosome, namely chromosome 12. Within these four aNEN, we observed possible evidence for extrachromosomal DNA due to copy-number oscillations between one low ($CN \leq 4$) and one very high ($CN \geq 10$) states, consistent with the presence of double minutes[35–37].

**Catalog of the cohort-wide mutational signatures provide biological insights into treatment effect.** Different mutational processes, such as exposure to exogenous or endogenous mutagens and defective DNA-repair mechanisms generate unique combinations of mutational trinucleotide contexts which are reflected in mutational signatures[38,39]. To determine these mutational signatures within aNEN, we performed de novo mutational signature analysis and determined the contribution of previously described SBS mutational signatures (COSMIC v3). The de novo mutational signature assessment revealed seven signatures, denoted as Sig. A to Sig. G, (Supplementary Fig. 6b, h, i) which all strongly correlated to previously known mutational signatures (Supplementary Fig. 6a–f). In particular, we observed samples with large relative contributions (>20%) of de novo signatures similar to the known signatures associated with aging (SBS1 & 5; Sig A and D), APOBEC activity (SBS2 & 13; Sig B.), tobacco smoking (SBS4; Sig. F.), alkylating agents exposure (SBS11; Sig E.), 5-Fluorouracil exposure (SBS17a-b; Sig. C.) and *MUTYH* mutations (SBS36; Sig. G.).

Overall, the mutational signature profiles do not differ greatly within the aNEN cohort. SBS5 ($n = 48$; putative clock-like), SBS8 ($n = 45$; possibly late-replication errors[40]), SBS40 ($n = 22$; Unknown), SBS3 ($n = 16$; HRD-like), SBS1 ($n = 10$; clock-like), SBS39 ($n = 7$; Unknown), and SBS9 ($n = 5$; polymerase η (*POLH*) activity) were classified as dominant signatures (i.e., contributed at least 10% of total contribution within ≥5 aNEN; Fig. 2e). When comparing between our major subgroups (aNEC, midgut- and pancreas-derived aNET), we observed significant ($q \leq 0.05$) differences for five previously described SBS mutational signatures (Supplementary Fig. 6g). The relative contribution of SBS3 (HRD-like) and SBS5 (clock-like) was lower in aNEC compared to midgut- and/or pancreas-derived aNET whilst conversely, SBS18 (reactive oxygen species) was elevated in aNEC. In addition, SBS8 (possibly late-replication errors) was elevated in midgut-derived aNET compared to the others. Finally, the relative presence SBS40 (unknown) was higher in pancreas-derived aNET compared to others.

Two included aNEC of unknown primary localization are characterized by high-TMB (≥10) and SBS4, which is associated with smoking; likely due to tobacco mutagens. This could reflect that these metastases could be primary lung non-small-cell lung cancer. However, as no somatic coding mutations in canonical lung cancer-associated genes were observed and the clinicopathological data of these patients did not point to any different primary tumor other than a NEC, it seems unlikely that these could be primary non-small-cell lung cancers. Smoking has also been implicated as a risk factor for pulmonary and extrapulmonary NEC such as those of the urinary bladder and the esophagus[41].

Strikingly, the only high-TMB (pancreatic) NET was strongly characterized by SBS11, which exhibits a mutational pattern resembling that of alkylating agents, with a strong enrichment for C/T (G > A) transitions. Previously, an association between treatment with the alkylating agent temozolomide and SBS11 mutations has been found[38,42]. This same patient showed the highest TMB with a TMB of 21.3 (median TMB of NET: 1.09) and was treated with a combination of 5-fluorouracil and streptozocin before undergoing a biopsy for the CPCT-02 study. Streptozocin is a capable of DNA alkylation and inhibition of DNA synthesis, and its mechanism of action closely resembles that of temozolomide.

One aNET was strongly characterized by SBS36, associated with base excision repair (BER) deficiency due to *MUTYH* alterations, C > A mutations and previously also seen in pancreatic NET[42–44]. Strikingly, this tumor did not harbor specific somatic alterations within *MUTYH* but possessed a heterozygous germline pathogenic missense mutation within *MUTYH* (c.527A>G / p.Tyr176Cys; rs34612342) coupled with a complete loss of a single chromosome 1, resulting in subsequent loss of heterozygosity.

**Driver catalog of aNEN.** We next performed an unbiased driver gene discovery analysis by performing GISTIC2[45] to detect recurrent somatic copy-number alterations and dN/dS[46] to detect genes under positive (or negative) selection pressure on the entire aNEN cohort and separately on all aNET and aNEC samples. With this analysis, we detected eighteen focal deletion peaks and two focal copy-number amplifications peaks throughout the genome ($q \leq 0.1$) and ten genes enriched with non-synonymous mutations ($q \leq 0.1$; Fig. 3 and Supplementary Fig. 7). Within these focal peaks, several oncogenes and tumor suppressors were present which could be the potential target of the copy-number alteration. These genes, which have been previously associated as driver genes in NET and/or pan-cancer cohorts[5,11,27], are shown in Fig. 3 for all aNEN with a distinction between aNEC and aNET. We detected several previously known tumor suppressors and oncogenes such as *TP53, KRAS, MEN1, RB1, CDKN1B, DAXX,* and *APC* enriched with non-synonymous mutations ($q \leq 0.05$) as well three additional genes (*LPCAT2, SETD2,* and *CREBBP*) just above the statistical threshold value ($q \leq 0.1$). By overlapping known drivers within the observed focal amplification and deletion peaks, we detected a plethora of putative drivers with copy-number alteration; such as deletions of *TP53, CDKN2A, CDKN2B, CDKN1B, PTPRD, DR1, CBFA2T3, PLCG2, ANKDR11, IRF8, LINC01881, PRKN, ZNF407,* common fragile sites such as *DMD, FHIT* and *MACROD2,* and amplifications of genes such as *PCAT1/MYC* and *MDM2.* Furthermore, focal deletions of additional genes such as *CAMTA1, DLUE1/2, TRIM13, KCNRG, FXD1* were found in ≤2 samples (Supplementary data 1). Large perturbations on chromosome 12q15 (*MDM2*) were observed within aNEN harboring chromothripsis (Supplementary Fig. 5). Furthermore, we could detect a single in-frame fusion of the common fusion-partner *EWSR1* seen in pNET[5]. Moreover, we observed only two genes harboring hotspot coding mutations (on base-level) which were shared between three samples (*ZNF829* and *KRAS*) and seven genes between two samples (*UHRF1BP1L, CDKN1B, MEN1, LEKR1, OR5L1, CTNNB1,* and *GNAS*; Supplementary data 1).

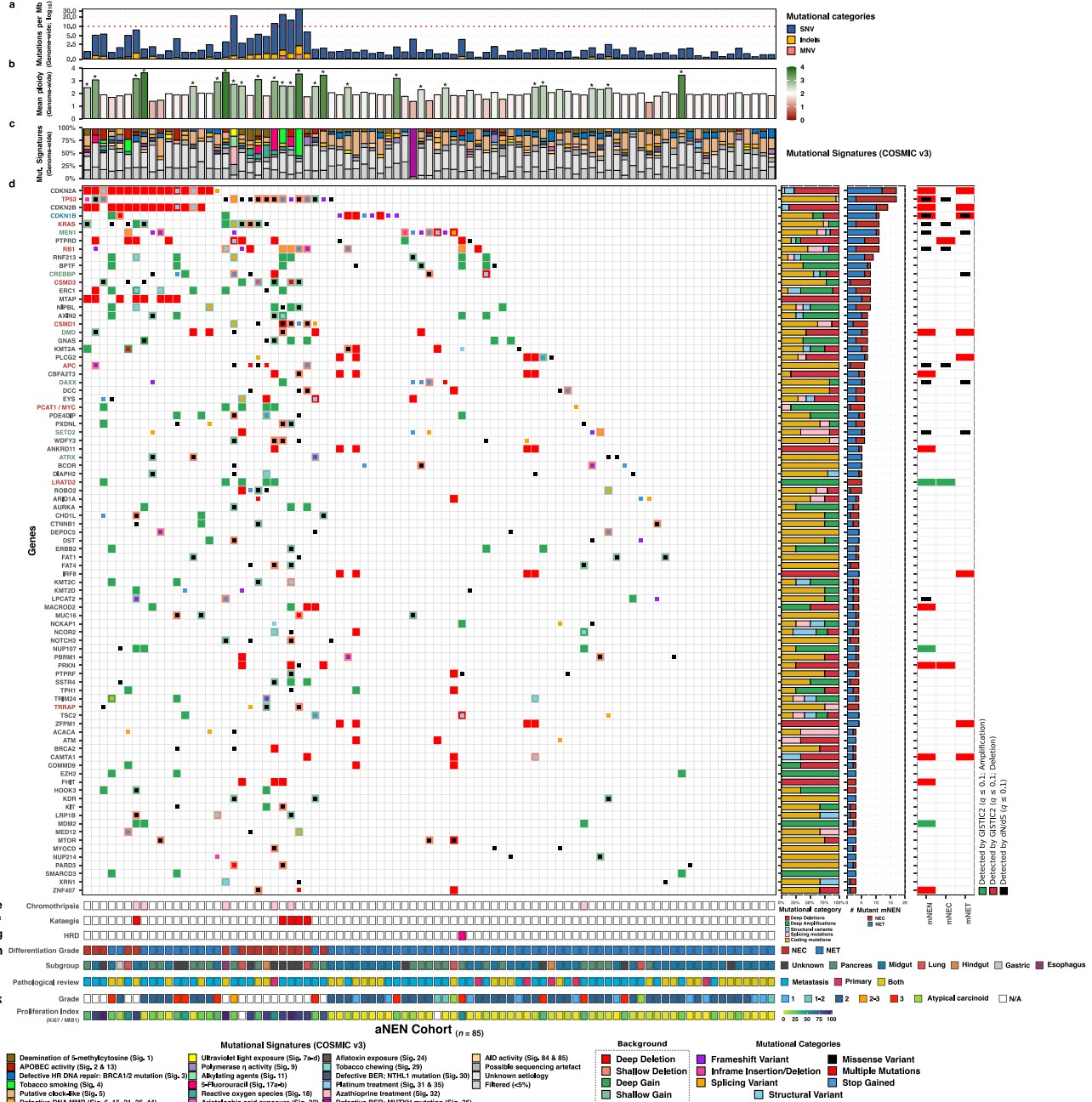

**Fig. 3 Putative drivers and NEN-associated genes within the aNEN cohort as detected by unbiased discovery (dN/dS, GISTIC2) and literature.**
Overview of putative drivers harboring coding mutations within ≥3 aNEN. We show putative drivers as detected by dN/dS and/or GISTIC2 and supplemented this list with additional NEN-associated drivers. aNEN and genes are sorted based on mutually exclusivity of the depicted putative drivers. In addition, genes found to be mutually exclusive between our major subgroups are highlighted in the respective color of the enriched subgroup; Supplementary Fig. 8e). This overview depicts the genomic features and the somatic inventory for the entire aNEN cohort (*n* = 85). **a** Number of genomic mutations per megabase over the entire genome (TMB). Threshold for high-TMB (≥10) is shown by a horizontal red dashed line. Y-axis is shown in log₁₀-scale. **b** Mean genome-wide ploidy, ranging from 0 (red) to 4 (green; tetraploid). Diploidy is shown in white. Suspected whole-genome duplication (WGD) events have been marked by an asterisk (*). **c** Relative contribution of the COSMIC single-base substitution mutational signatures (v3; *n* = 67). Proposed etiology of signatures is denoted below. **d** Overview of coding mutation(s) per aNEN, (light-)green or (light-)red backgrounds depict copy-number aberrations whilst the inner square depicts the type of (coding) mutation(s). The adjacent bar plots represent the relative proportions of mutational categories per gene, the percentage of aNEC (in red) and aNET in blue harboring mutation and the dN/dS and/or GISTIC2 support, per analysis. **e** Presence of chromothripsis; aNEN with chromothripsis are shown in pink. **f** Presence of kataegis; aNEN with ≥1 kataegis events are shown in red. **g** Status of homologous recombination deficiency (HRD), as determined by CHORD; aNEN with *BRCA1/2*-associated HRD are shown in pink. **h** Differentiation grade of the aNEN; aNEC in red, aNET in blue. **i** Primary localization of the aNEN. **j** Origin of the respective pathological record used to determine differentiation grade and proliferation index. **k** Differentiation grade based on the pathological record: 1 (sky-blue), 1-2 (teal), 2 (dark blue), 2-3 (orange), 3 (red), atypical carcinoid (green) and not available (N/A; white). **l** Proliferation index (KI67/MIB-1) from 0 to 100 based on the pathological record.

We observed an overall heterogeneous pattern of putative drivers, the most frequently putative driver was found to be *CDKN2A/B* ($n = 17$; 14), followed by *TP53* ($n = 17$), *CDKN1B* ($n = 11$), *PTPRD* ($n = 11$), *KRAS* ($n = 11$), *MEN1* ($n = 11$) and *RB1* ($n = 11$). Strikingly, a significant portion of the total aNEN cohort had no mutual putative driver(s) (9 out of 85; 11%) and only contained patient-specific putative drivers.

We next investigated whether any form of mutational enrichment, such as somatic alterations within certain genes (mutations and/or copy-number alterations) or evidence of large-scale events (kataegis and chromothripsis), could be related to one of our three major subgroups relating to subtype or primary localization; being aNEC ($n = 16$), pancreas- ($n = 20$), and midgut-derived aNET ($n = 39$). Using a one-sided Fisher's exact test (with Benjamini–Hochberg correction) on relevant genes ($n = 20$) captured within either within our dN/dS and GISTIC2 analysis or present as mutant genes with either mutations or copy-number alterations within 20% of each major subgroup, we detected the enrichment of at least one such event(s) within these subgroups (Supplementary Fig. 8e).

Within aNEC, an enrichment of alterations within *TP53* (88% of aNEC), *KRAS* (50%), *RB1* (50%), *CSMD3* (44%), *APC* (31%), *CSMD1* (31%), *LRATD2* (31%), PCAT1/*MYC* (31%), *TRRAP* (25%), and presence of kataegis (31%) and chromothripsis (25%) could be appreciated ($q \le 0.05$). Likewise, within pancreas-derived aNET, an enrichment was seen for *MEN1* (40% of pancreas-derived aNET), *DAXX* (25%), *DMD* (25%), *SETD2* (25%), *ATRX* (20%) and *CREBBP* (20%) whilst midgut-derived aNET revealed enrichment of *CDKN1B* alterations (23% of midgut-derived aNET).

**Genomic differences relating to primary localization of aNET.** Due to distinct prognosis and previous genetic associations, we investigated genome-wide differences in regards to primary localization within the aNET population ($n = 69$). We observed several genome-wide differences relating to primary localization (Fig. 2, Supplementary Fig. 8), such as the median genome-wide TMB; ranging from 1.05 (aNET-Midgut; $Q_1–Q_3$: 0.75–1.4) and 1.07 (aNET - Unknown; $Q_1–Q_3$: 0.84–1.53) to 1.27 (aNET - Other; $Q_1–Q_3$: 1.10–1.44) and 1.35 (aNET-Pancreas; $Q_1–Q_3$: 0.9–2.12). A similar pattern was detected regarding the number of distinct genes with coding mutations. Midgut-derived aNET also presented a surprisingly low number of SVs compared to the other aNET subpopulations.

Next, we investigated possible differences in putative drivers between our major aNET subpopulations, being midgut- ($n = 39$) and pancreas-derived ($n = 20$) aNET (Fig. 4, Supplementary Fig. 8). The copy-number profiles (GISTIC2) of both populations differed, in which midgut-derived aNET presented focal deletion peaks at 9p21 (*CDKN2A/B*), 11q23 (7 possible driver genes), 12p13 (*CDKN1B*), 13q14 (17 genes), 14q24 (20 genes) and 16q23 (5 possible driver genes; common fragile site) coupled with an overall flat diploid profile. Pancreas-derived aNET presented a different profile harboring focal deletion peaks at 2q37 (*LINC01881*), 9p21 (*CDKN2A/B*) and Xp21 (*DMD*; common fragile site gene) couples with a more instable genomic profile, including several samples with large-scale chromosomal losses (Supplementary Figs. 7 and. 8c). When investigating the statistically significant large-scale copy-number alterations of the chromosomal arms, we also detect striking differences between the major subgroups (Supplementary Fig. 9). Within aNEC, we detected a large number of samples (69%) harboring a loss of $22_q$. Midgut-derived aNET revealed amplifications of chromosome $4_{p/q}$, $5_{p/q}$, $7_{p/q}$, $10_{p/q}$, $14_{p/q}$, $20_{p/q}$ and loss of $9_{p/q}$ in various samples (~30%) and a loss of $18_{p/q}$ in 66% of samples.

This re-confirms the high frequency of chromosome 18 loss in midgut-derived NET and the association with *DDC*[47], as *DCC* ($18_{q21.2}$) is the most recurrently mutated gene on chromosome 18 in our cohort also ($n = 6$) together with *CDH7* ($n = 6$; $18_{q22.1}$). Finally, over half of pancreas-derived aNET revealed amplifications of chromosome $5_{p/q}$, $7_{p/q}$, $9_q$, $12_{p/q}$, $13_q$, $14_{p/q}$, $17_{p/q}$, $18_{p/q}$, $19_{p/q}$, $20_{p/q}$ and loss of $22_q$.

Unbiased driver gene analysis (dN/dS) on midgut-derived aNET presented *CDKN1B* whilst pancreas-derived aNET revealed *MEN1*, *DAXX*, and *SETD2*. Several genes (present in $\ge 2$ samples) were found only, or predominately, within midgut-derived aNET: *CDKN1B*, *KMT2A*, *PSIP1*, and *PTPRD* (Fig. 4; Supplementary Fig. 8e). Conversely, *MEN1*, *DAXX*, *DMD*, *SETD2*, *ATRX*, *CREBBP*, *DST*, *KDR*, *PTPRC*, and *TSC2* were found to be mutated only within pancreas-derived aNET. Several midgut-derived aNET ($n = 9$; 23%) did not readily present a shared mutual driver and only harbored somatic mutations in private or as-of-yet unassociated cancer driver genes.

**Clinically actionable mutations.** We observed forty-two aNEN (49%) harboring one or more target-specific or general somatic aberrations which are known as possible (and responsive) druggable targets against currently available (or under development) treatment agents. Twenty-one aNEN (24%) harbored somatic aberrations corresponding to a treatment that is currently registered for NEN or specifically for the NEN subtype of that particular patient (Fig. 5, Supplementary data 1). In addition, 14 patients (16%) could benefit from therapies that are off-label, but are commonly considered best practice for NEN. Another seven patients (8%) could benefit from drugs which are registered for another indication but not currently administered in NEN treatment. We found *RB1* ($n = 11$), *KRAS* ($n = 11$), *MTAP* ($n = 8$), high-TMB ($\ge 10$; $n = 5$), *RICTOR* ($n = 4$), and *TP53* ($n = 4$) to be the most frequently observed (target-specific or general) somatic aberrations which granted eligibility to various possible treatment options. In total, 10 midgut-derived aNET (26%) and 11 pancreas-derived aNET (55%) revealed potentially responsive alterations in various genes and most strikingly, almost all aNEC (94%) revealed potential responsive targets due to *RB1* and/or *KRAS* mutations or toward checkpoint inhibitors due to high TMB ($\ge 10$).

**Discussion**

Historically, NEN has long been considered as a difficult malignancy to diagnose, monitor, and treat due to presentation of an inherently wide spectrum of disease progression, cellular differentiation and low mutational burden, resulting in few targetable mutations and a relatively stable tumor genome. Indeed, aNET is characterized by the lowest TMB of all metastatic cohorts sequenced in the CPCT-02 study[27]. This study is the first to have an in-depth look into the whole genome and mutations of a large cohort of 85 advanced NEN from various primary localizations and differentiation grades. The relatively large number of unknown primary tumor localizations in this aNEN cohort ($n = 12$; 14%) reflects the difficulties in daily clinical practice to determine the site of origin for aNEN. Recently, we have become more aware of the phenomena of trans-differentiation, in which a NEC arises within a pre-existing adenocarcinoma of for instance the lung or prostate. However, in the six aNEC patients with an unknown primary tumor, no molecular clues, such as *TMPRSS2-ERG* fusions were found pointing to a specific tissue of origin.

In our aNEN cohort, it is apparent that the molecular landscape of aNEC is markedly dissimilar from that of the more differentiated aNET, in terms of mutational burden (median TMB of 5.45 vs. 1.09, respectively), genomic stability, and distinct

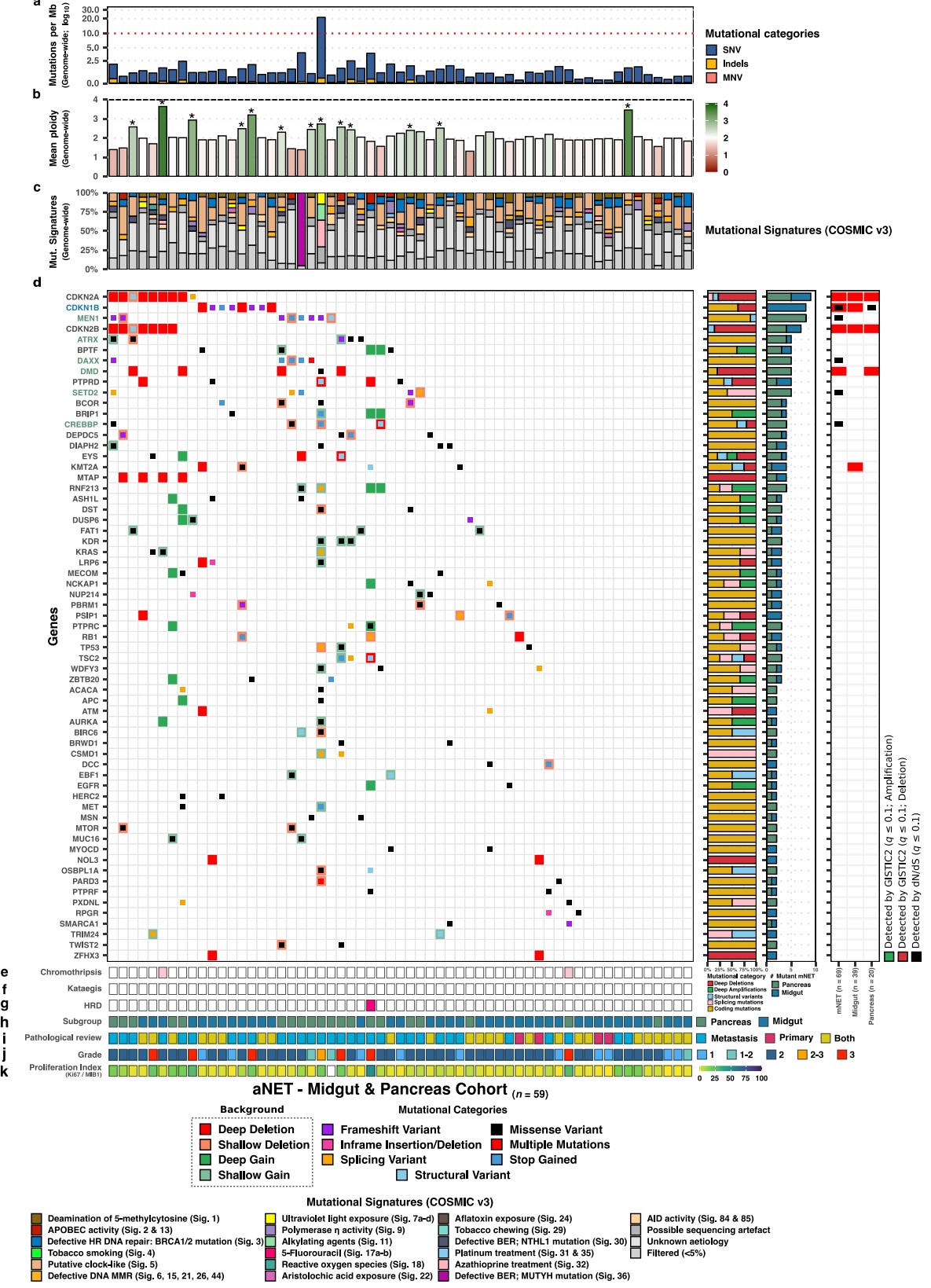

**Fig. 4 Putative drivers and NEN-associated genes within the pancreas- and midgut-derived aNET as detected by unbiased discovery (dN/dS, GISTIC2) and literature.** Overview of putative drivers harboring coding mutations within at least two pancreas- and/or midgut-derived aNET. We show putative drivers as detected by subgroup-specific dN/dS and/or GISTIC2 and supplemented this list with additional NEN-associated drivers. aNET and genes are sorted based on mutually exclusivity of the depicted putative drivers. Same layout as Fig. 3, except the adjacent middle-outer bar (in **d**) depicts the percentage of pancreas-derived m(NET) in green and midgut-derived aNET in blue. In addition, genes found to be mutually exclusive between our major subgroups are highlighted in the respective color of the enriched subgroup (aNET-Pancreas (green and aNET-Midgut (blue); Supplementary Fig. 8e).

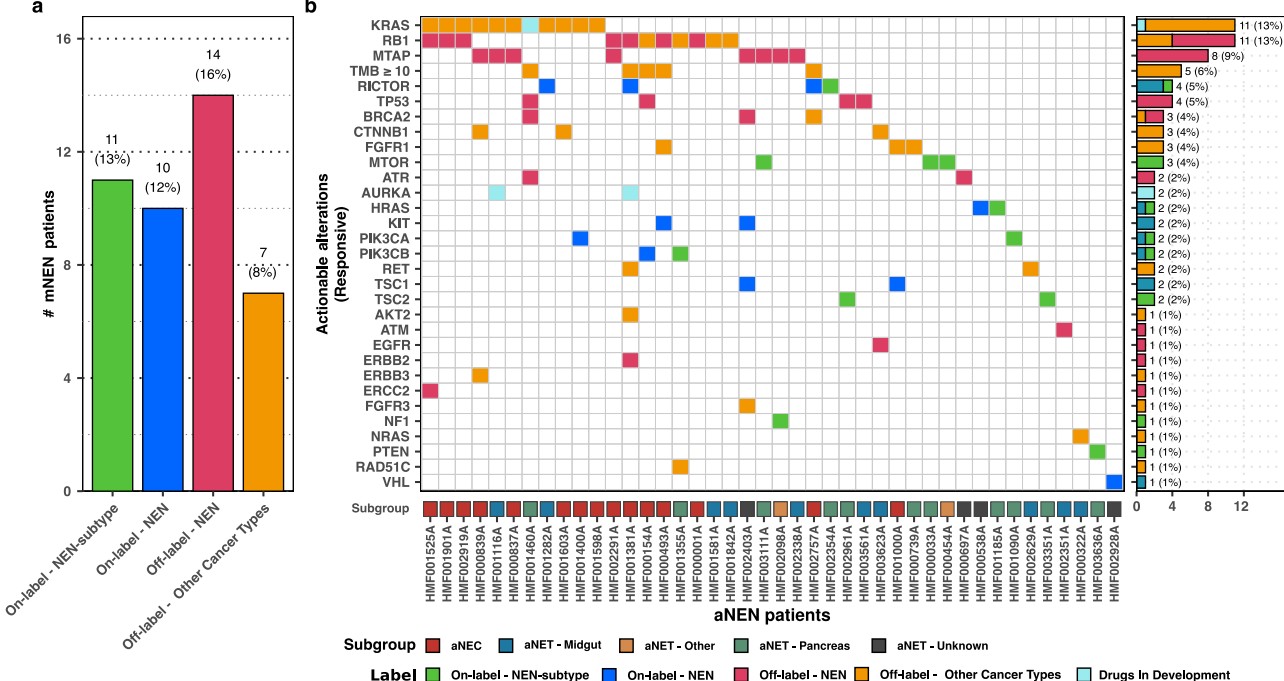

**Fig. 5 Clinically actionable somatic alterations observed within aNEN. a** Overview of distinct aNEN harboring current clinically actionable alterations for on- and off-label NEN therapies. The highest NEN-therapy option (ranked as on-label NEN subtype (green), on-label NEN (dark blue), off-label for NEN (pink), off-label for other cancer types but currently available (orange) and drugs in development (turquoise) per distinct aNEN is shown. **b** aNEN harboring current clinically actionable alterations, per gene. Full description: aNEN harboring current clinically actionable alterations, per gene. The highest NET-therapy option per aNEN and gene is shown. Bottom track represents the categorized primary localization of the aNEN (aNEC in red, midgut-derived aNET in blue, non-midgut/pancreas-derived aNET in orange, pancreas-derived aNET in green and aNET of unknown origin in black) whilst the right-hand side figure shown the number of samples harboring a somatic alteration within the given gene and the proposed level of therapy.

mutant (driver) genes. With respect to TMB, four aNEC and a single aNET presented a high-TMB genotype (TMB ≥ 10) which could render these patients eligible for immune-based therapies such as checkpoint inhibitors[48,49].

The single high-TMB pancreas-derived aNET presented a striking contribution of the mutational signature associated with alkylating agents (temozolomide) and was previously treated with a combination of 5-fluorouracil and the alkylating antineoplastic agent streptozocin. The mechanism of action for streptozocin closely resembles that of temozolomide as both react with DNA by undergoing substitution reactions forming a methyldiazonium ion, resulting in methylation of primarily N$^7$ guanine (67%). They both induce high levels of DNA methylation, and recognition and repair of this methylation results in single- and double-strand DNA breaks[50]. To the best of our knowledge, no data have been published on a correlation between hypermutation and strepto-zocin treatment, but as streptozocin and temozolomide so closely resemble each other in their mechanism of action, one can hypothesize the same mechanism to occur in streptozocin-treated patients. It would be interesting to investigate whether prior treatment with streptozocin or temozolomide indeed induces high-TMB in aNEN, and if so, whether pre-treatment with streptozocin or temozolomide could render these tumors more sensitive to checkpoint inhibition. Likewise, temozolomide (with capecitabine) for advanced pancreatic NETs has shown to be an effective therapy for these patients[51]. Similarly, we observed a large contribution of the mutational signature associated with BER deficiency due to *MUTYH* aberrations in the second highest-TMB aNET, and indeed this patient harbored a pathogenic germline *MUTYH* allele coupled with a complete somatic loss of the respective chromosomal arm. *MUTYH* abnormalities have also previously described to occur in pancreatic NET[5]. A single

aNET presented a *BRCA2*-genotype associated with HRD but did not harbor (somatic) mutations within *BRCA2*. It did harbor a somatic mutation in *RAD51C*, a gene known to be involved with homologous recombination and repair of DNA.

Concerning genomic stability, we observed evidence of chromothripsis, a large-scale and catastrophic chromosomal rearrangement, within six aNEN (four aNEC, two aNET). Strikingly, four out of six chromothripsis events occurred on chromosome 12. In addition, we observe the first occurrence of localized hypermutation (kataegis) in five aNEC. Kataegis encompasses a pattern of localized hypermutations, which has been identified in various, but not all and to a varying degree, cancer types[52,53]. These regions of kataegis often co-localize with regions of genetic rearrangements. Kataegis is thought to arise from frequent genomic C-to-U deamination events as a result of APOBEC-family enzyme activity, a DNA cytosine deaminase which was recently identified as an internal and thus far unrecognized source of DNA damage and mutagenesis in various cancer types[54]. More recently, kataegis, rather than TMB, microsatellite instability or mismatch repair deficiency, was found to independently correlate with PD-L1/PD-L2 expression, and could thus be a marker in response to immune checkpoint inhibition[55].

Using unbiased driver gene analysis (dN/dS and GISTIC2) on the aNEN cohort, and on aNEC/aNEC separately to explore putative driver genes, we (re-)discovered 10 genes to be enriched with non-synonymous mutations (*TP53, CDKN1B, KRAS, MEN1, RB1, CREBBP, APC, DAXX, LPCAT2*, and *SETD2*) and detected 18 focal deletion and 2 focal amplification peaks overlapping with a plethora of (driver) genes, including deletions of *TP53, CDKN2A, CDKN2B, CDKN1B, PTPRD, CBFA2T3, CAMTA1, ANKDR11, LINC00881, PRKN, ZNF407* and fragile site genes *FHIT, DMD* and *MACROD2*, and amplifications of *PCAT1/MYC*

and *MDM2*. Investigation of mutational enrichment within our major subgroups revealed that somatic alterations in *TP53*, *KRAS*, *RB1*, *CSMD3*, *CSMD1*, *MYC*, *APC*, *LRATD2*, and *TRRAP*, as well as the presence of chromothripsis and kataegis was enriched within aNEC. Within pancreas-derived aNET, we report the enriched presence of mutant *MEN1*, *DAXX*, *DMD*, *SETD2*, *ATRX*, and *CREBBP*, whilst midgut-derived aNET showed preference for *CDKN1B* alterations.

As previously mentioned, the majority of these detected somatic aberrations have been previously associated to primary NEN in regards to their tissue of origin. These include the associations with midgut-derived NET (*CDKN1B*)[10,13], lung NET (*FHIT*)[56–58], pNET (*TP53*, *MEN1*, *ATRX*, *DAXX* and *SETD2*)[3–5,7,15] and NEC (*TP53, KRAS, MYC, APC* and *RB1*, and chromothripsis)[17,18,59,60]. Aberrations within *CDKN2A* and *CDKN2B* have been associated to gastro-intestinal NETs[61,62] and have been observed with increased mutational frequency within metastatic pNET compared to primary pNET and is associated to poor prognosis[63].

A recent large-scale study utilizing organoids derived from gastroenteropancreatic neuroendocrine (GEP) neoplasms also revealed similar genomic landscapes and (mutually exclusive) enrichment for drivers such as *TP53*, *RB1*, *APC*, and *MYC* within NEC for GEP-NEN organoids and chromosome-wide loss of heterozygosity within both NET and NEC tissues[64]. Concordantly, mutational enrichment of drivers within one population (i.e. pancreas-derived NET) does not imply exclusivity; e.g., *MEN1* aberrations were also found to be (sporadically) present within GEP-NECs and within a single NEC of our cohort.

Other frequently altered genes within our aNEN cohort are associated with various other malignancies (*PTPRD*[56], *CBFA2T3*[65], *ANKRD11*[66–68], and *MDM2*[69]) or genomic instability (*DMD* and *PRKN*[57], *MACROD2*[70]). In particular, *CSMD1* and *CSMD3* (CUB And Sushi Multiple Domains 1 and 3) were found almost exclusively mutated within aNEC (31% and 44% of aNEC, respectively) yet have not previously obtained much attention in context to aNEC. *CSMD1*, a regulator of complement activation and inflammation, has been proposed as a tumor suppressor gene in advanced oral, gastric, prostate and breast cancer and subsequent loss of *CSMD1* functionality is associated to poor prognosis and enhanced proliferation, migration and invasion[71–74]. Moreover, *CSMD3* is reported as frequently mutated in lung cancers and associated with proliferation of airway epithelial cells[75] and has been recently also reported as enriched within NEC compared to NET[76]. Taken together, this prompts further investigation for *CSMD1* and *CSMD3* as aNEC-related drivers.

Currently, the choice of treatment in an individual aNEN patient is, apart from factors such as comorbidity and patient preference, determined by primary tumor localization, proliferation index (as determined by Ki-67 or MIB-1 staining), and somatostatin expression. The distinction based on primary tumor localization stems from the different embryologic structures the tumor can originate from, e.g. foregut, midgut or hindgut. When comparing the various origins of the aNEN at a genomic level, we conclude that aNEN harbors a strikingly low TMB compared to cancers[29], yet do observe slight deviations on total TMB; ranging from 1.05 (aNET-midgut; $Q_1$-$Q_3$: 0.75–1.4) and 1.07 (aNET—unknown; $Q_1$-$Q_3$: 0.84–1.53) to 1.27 (aNET—other; $Q_1$-$Q_3$: 1.10–1.44) and 1.35 (aNET-Pancreas; $Q_1$-$Q_3$: 0.9–2.12) to 5.45 (aNEC; $Q$-$Q_3$: 3.8–8.85). In addition, when we compared the two largest groups of aNET per primary localization (midgut and pancreas), we can readily distinguish between the two subtypes based on somatic mutation and copy-number profiles. Yet strikingly, many midgut-derived aNET ($n = 9$; 23%) did not present a mutual driver gene but each was characterized by

distinct sets of mutated genes reflecting the heterogenous nature of the malignancy.

Almost half of aNEN ($n = 42$; 49%) harbored a specific genomic alteration or genotype for which an FDA-approved drug is currently available, either on (registered for that indication) or off label. Thus, WEG revealed 49% of aNEN patients harboring clinically relevant and potentially targetable somatic aberrations which could possibly extend their treatment repertoire. It should be noted that we do not yet know whether these identified associations between genomic alterations and specific drugs indeed translate into clinical response in these patients. However, for instance, when looking at TMB as a predictive factor for checkpoint inhibitors, it was recently shown that TMB-high aNEC can respond to pembrolizumab[77]. These drugs are currently not readily available for these patients, but could provide new treatment options in the future. When deciding upon a new line of systemic treatment, a metastatic biopsy could always be considered, preferably in the context of a study, as this could shed light upon additional and effective treatment options for these late-stage patients with otherwise few remaining treatment options. In the Netherlands, we have the DRUP study active, a study in which patients for whom no standard treatments are currently available and whom might be treated with anticancer treatments outside of their approved label based on the presence of actionable mutations in their tumors[28].

In this current study exploring the largest whole-genome sequenced aNEN repository to date ($n = 85$), we focused on the genetic aberrations driving aNEN and analyzed several additional aspects of genomic instability, such as SVs, kataegis, chromothripsis, and HRD. This study improves our understanding of the complex molecular makeup of (m)NEN and reveals that the underlying genomic alterations could be exploited for better distinction of tumor subgroups and new treatment options. This study furthermore underscores that whilst the number of genetic aberrations is increased[27], the inventory of somatic drivers does not significantly change between primary and metastatic NEN. The major advantages of characterizing the genomic landscape of metastatic NEN lie within the identification of potentially actionable targets and treatment-induced (resistance-)mechanisms within the late-stage disease.

In addition, the recent major collaborative efforts in acquiring, (whole-genome) sequencing and releasing several large-scale pan-cancer datasets comprising both primary and metastatic malignancies, such as the PCAWG[78] and CPCT-02[27], could spark insights and the development of methods on how to fully interrogate and map the whole tumor genome, including the still relatively unexplored non-coding regions. This could deduce new shared oncogenic mechanisms but also, by contrast, reveal driving forces unique to (m)NEN. Within this presented aNEN repository, the full range of the somatic principles driving this enigmatic disease are likely still hidden from us but ever-present.

## Methods

**Patient cohort and study procedures**. Patients with aNEN were recruited under the study protocol (CPCT-02 Biopsy Protocol, ClinicalTrial.gov no. NCT01855477; Suppl. Note 1) of the Center for Personalized Cancer Treatment (CPCT) within the CPCT-02 and the DRUP (The Drug Rediscovery Protocol (DRUP Trial), ClinicalTrial.gov no. NCT02925234) studies. All analyzed biopsies were taken prior to treatment within the DRUP trial. The CPCT-02 (NCT01855477) and DRUP (NCT02925234) clinical studies were approved by the medical ethical committees of the University Medical Center Utrecht and the Netherlands Cancer Institute, respectively. Patients were eligible for inclusion if the following criteria were met: (1) age ≥18 years; (2) locally advanced or metastatic solid tumor; (3) indication for new line of systemic treatment with registered anticancer agents; (4) safe biopsy according to the intervening physician. All patients have given explicit consent for WEG and data sharing for cancer research purposes. The study procedures consisted of the collection of matched peripheral blood samples for reference DNA and image-guided percutaneous biopsy of the preferred metastatic site or, if no

high-quality metastatic biopsy was available, a biopsy of the primary tumor site was collected. For the current study, patients were included for biopsy between May 10, 2016 and July 17, 2018 resulting in a cohort of 85 distinct patients from 13 Dutch hospitals (Supplementary data 1).

**Collection of the pathological records and generalization of pre-treatment(s).** Primary tumor characteristics of the 85 included aNEN patients were checked within the nationwide network and registry of histo- and cytopathology in the Netherlands (PALGA)[79].

From PALGA, we collected the differentiation grade and proliferation index (Ki67/MIB-1) based on the pathological records of the patient-specific primary and/or any metastatic lesion. If more than one pathological report was available, we chose to include the report most close in date, but always prior to, the biopsy for the CPCT-02 study.

The pre-treatment(s) of aNEN patients prior to the collection and sequencing of the tumor biopsy has been collected and generalized on treatment classification. Out of all included aNEN patients ($n = 85$), 26 patients received pre-treatment according to our clinical records.

**Collection, sequencing, and processing of aNEN biopsies.** Blood samples were collected in CellSave preservative tubes (Menarini-Silicon Biosystems, Huntington Valley, PA, USA) and shipped by room temperature to the central sequencing facility at the Hartwig Medical Foundation. Tumor samples were fresh-frozen in liquid nitrogen directly after the procedure and send to a central pathology tissue facility. Tumor cellularity was estimated by assessing a hematoxylin-eosin stained 6-micron section. Subsequently, 25 sections of 20 microns were collected for DNA isolation. DNA was isolated with an automated workflow (QiaSymphony) using the DSP DNA Midi kit for blood and QiaSymphony DSP DNA Mini kit for tumor samples according to the manufacturer's protocol (Qiagen). DNA concentration was measured by Qubit™ fluorometric quantitation (Invitrogen, Life Technologies, Carlsbad, CA, USA). DNA libraries for Illumina sequencing were generated from 50–100 ng of genomic DNA using standard protocols (Illumina, San Diego, CA, USA) and subsequently whole-genome sequenced in a HiSeq X Ten system using the paired-end sequencing protocol ($2 \times 150$bp) for both the biopsy and matched blood sample.

Subsequent alignment, somatic mutation detection, and in silico tumor cell percentage estimation were performed in a uniform manner as detailed by Priestley et al.[27]. Briefly, paired-end sequencing reads were aligned against the human reference genome (GRCh37) using BWA-mem (v0.7.5a)[80]. Duplicate reads were marked and small insertion and deletions (InDels) were realigned using GATK IndelRealigner (v3.4.46). Prior to somatic SNV and InDel variant calling, base qualities were recalibrated using GATK BQSR (v3.4.46)[81]. Somatic SNV, InDels, and MNV were called by Strelka (v1.0.14) using the matched peripheral blood WGS sample for matched-normal variant calling[82].

Additional in-depth settings and optimizations of the HMF pipeline are described by Priestley et al.[27] and tools are available at https://github.com/hartwigmedical/.

The somatic mutations (SNV, InDels, and MNV) were further annotated with Ensembl Variant Effect Predictor[83] (VEP, version 99, cache 99_GRCh37) using GENCODE (v33) annotations in tandem with the dbNSFP[84] plugin (version 3.5, hg19) for gnomAD[85] population frequencies. SIFT[86] and PolyPhen-2[87] scoring was applied for additional functional effect prediction.

During downstream analysis, we only retained SNV, InDels, and MNV which passed all of the following heuristic filters; default Strelka filters (PASS-only), gnomAD exome (ALL) allele frequency <0.001, gnomAD genome (ALL) <0.005, not present in ≥5 samples from the Hartwig Medical Foundation germline panel-of-normals (GATK Haplotyper) and not present in ≥3 samples from the Hartwig Medical Foundation Strelka-specific somatic blacklist.

Putative protein-altering (coding) or high-impact (e.g., splicing) mutations were aggregated per sample and gene by selecting the most deleterious annotated effect (from VEP) on any known overlapping gene-wise transcript (except those transcripts flagged as retained intron and nonsense-mediated decay). In addition, SVs with a Tumor Allele Frequency (TAF) ≥ 0.1, as calculated by PURPLE and GRIDSS[88], that overlapped only partly with the respective coding sequences (i.e., not all exons of the respective gene), were annotated as 'SV' mutations. Multiple coding mutations and/or SV per gene were annotated as 'multiple mutations'.

Discovery of somatic SVs, copy-number alterations, and in-frame fusions of EWSR1 was performed using the GRIDDS (v2.9.3), PURPLE (v2.47) and LINX (v2.47) suite[88]. During the downstream analyses, we only retained somatic SVs passing all default QC filters (PASS-only) and with an upstream and/or downstream TAF ≥ 0.1.

Mean read coverages of the reference and tumor samples were calculated using Picard Tools (v1.141; CollectWgsMetrics) based on GRCh37[89]. Genomic and coding TMB was calculated as previously described by van Dessel et al. (2019)[90]. Briefly, the number of somatic mutations (SNVs, InDels and MNVs) was divided over the total mappable bases and the superset of coding sequences, respectively.

**Discovery of genes under evolutionary selection.** We performed a dN/dS analysis on somatic mutations (SNV and InDels) using dndscv[46] (v0.0.1.0) on

respective genome sequences and transcript annotations using a custom transcript database based on ENSEMBL[91] Genes (v99)/GENCODE (v33) annotations. We performed a dN/dS analysis over the entire NEN cohort ($n = 85$) and four separate dN/dS analysis on the major subgroups (aNEC; $n = 16$, NET; $n = 69$, aNET-midgut; $n = 39$ and aNET-pancreas; $n = 20$). Genes-of-interest were selected based on the statistical significance, corrected for multiple hypothesis testing (Benjamini-Hochberg), which integrated all mutation types (missense, nonsense, essential splice-site mutations and InDels; qglobal_cv ≤ 0.1) and/or without InDels (qallsubs_cv ≤ 0.1).

**Detection and annotation of recurrent copy-number alterations.** To detect recurrent copy-number alterations, we performed a GISTIC2[45] (v2.0.23) analysis over the entire aNEN cohort and, again, four separate GISTIC2 analysis on the major subgroups (aNEC, aNET and pancreas- and midgut-derived aNET).

GISTIC2 was performed using the following settings:

gistic2 -b <inputFolder> -seg <inputSegmentation> -refgene hg19.UCSC. add_miR.140312.refgene.mat -genegistic 1 -gcm extreme -maxseg 4000 -broad 1 -brlen 0.98 -conf 0.95 -rx 0 -cap 3 -saveseg 0 -armpeel 1 -smallmem 0 -res 0.01 -ta 0.1 -td 0.1 -savedata 0 -savegene 1 -qvt 0.1.

Genes were annotated to GISTIC2 peaks ($q \leq 0.1$) based on the following strategy;

(1) GISTIC2 focal peaks (all_lesions.conf_95.txt) were overlapped to genes (from verified and manually annotated loci, no pseudogenes and read-throughs and from standard chromosomes; $n = 36574$) from GENCODE (GRCh37; v33), taking into consideration only the genes overlapping with at least 100 base pairs within the detected GISTIC2 peak.

(2) If a GISTIC2 focal peak overlapped with multiple GENCODE genes, a combined database containing known drivers detected in a metastatic pan-cancer dataset (CPCT-02)[27], COSMIC Cancer Gene Census (v85)[92], OncoKB Cancer Gene Census (June 2019)[93] Martincorena et al.[46], and Priestley et al.[27] were used to further pinpoint the possible target gene(s) ($n = 1272$), e.g., if a GISTIC2 peak overlapped both PTEN and near-adjacent non-driver gene, only PTEN would be chosen as possible gene. The list of all overlapping GENCODE[94] (v33) genes per GISTIC2 peak can be found in Supplementary data 1.

(3) If no overlapping genes were found, GISTIC2 peaks were annotated with the nearest GENCODE (v33) protein-coding gene ($n = 19,988$).

Genes detected as deep amplifications or deep deletions within GISTIC2 focal peaks were considered as GISTIC2-derived driver genes in this cohort.

**Mutational signature analysis.** Mutational signatures based on the trinucleotide contexts of SNVs were performed, using the MutationalPatterns package (1.10.0)[95] and as previously described[90]. The 96 SBS mutational signatures (COSMIC v3) as established by Alexandrov et al. (2019)[42], (matrix S$ij$; $i = 96$; number of trinucleotide motifs; $j =$ number of signatures) were downloaded from COSMIC (as deposited on May 2019). The proposed etiology of each SBS signature was derived from Alexandrov et al. (2019)[29], Petljak et al.[42], Angus et al.[19] and Christensen et al. (2019)[96].

In addition, de novo mutational signature analysis by MutationalPatterns was performed based on the max. number of relevant signatures as assessed using the NMF R package[97] (v0.21.0) with 1000 iterations (Supplementary Fig. 6d). By comparing the cophenetic correlation coefficient, residual sum of squares and silhouette, we opted to generate seven custom de novo signatures. Custom signatures were correlated to existing (COSMIC v3) mutational signatures using cosine similarity.

Per sample, mutational signatures with less than five percent relative contribution were categorized into the "Filtered (<5%)" category.

**Detection of chromothripsis.** Shatterseek[35] (v0.4) using default parameters was used to detect chromothripsis-like events. As input, we used the rounded absolute copy numbers (as derived by PURPLE) and SVs with an TAF ≥ 0.1 at either end of the breakpoint. The male sex chromosome (chrY) was excluded. The criteria for a chromothripsis-like event were based on the following criteria: (a) total number of intra-chromosomal SVs involved in the event ≥25; (b) max. number of oscillating CN segments (2 states) ≥7 or max. number of oscillating CN segments (3 states) ≥14; (c) total size of chromothripsis event ≥20 megabase pairs (Mbp); (d) satisfying the test of equal distribution of SV types ($p > 0.05$); and (e) satisfying the test of non-random SV distribution within the cluster region or chromosome ($p \leq 0.05$).

**Classification of homologous recombination deficiency genotypes.** To determine HRD due to possible loss of function of BRCA1 and/or BRCA2 (amongst others), we utilized the Classifier for HRD with default settings (CHORD; v2.0). CHORD uses a random-forest approach to classify samples into HR-deficient/HR-proficient categories[31]. Briefly, we make use of CHORD;[31] a random-forest-based classifier designed to classify samples with evidence of HRD (BRCA1-type, BRCA2-type or otherwise) by using all the information captured within all the somatic small mutations and somatic SVs of whole-genome sequenced samples. If a sample contains sufficient HRD-related genomic scars (SVs) and additional markers for HRD, that sample will be classified as HR-deficient (HRD).

**Detecting enrichment of mutant genes within major subgroups**. To determine the enrichment of mutant genes within our major subgroups (aNEC, midgut- and pancreas-derived aNET), we generated a list of potential driver genes based on captured genes through our dN/dS ($q \leq 0.1$) analysis and/or present within the focal amplification and deletion peaks captured by GISTIC2. We extended this list by selecting genes which contained a coding mutation in ≥20% of a respective subgroup or which harbored a deep amplification or deletion in ≥20% of the respective subgroup (i.e., 20% of the respective subgroup contained coding mutations and/or ≥20% contained a copy-number alteration, irrespective of coding mutation). Using this list of genes ($n = 20$), we performed a one-sided (enrichment) Fisher's exact test with Benjamini–Hochberg correction between each pairwise comparison per major subgroup against the remaining major subgroups (e.g., aNEC vs. the combined group of midgut- and pancreas-derived aNET).

**Inventory of clinically actionable somatic alterations and putative therapeutic targets**. Current clinical relevance of somatic alterations in relation to putative treatment options or resistance mechanisms and trial eligibility was determined based upon the following databases; CiViC[98] (Nov. 2018), OncoKB[93] (Nov. 2018), CGI[99] (Nov. 2018) and the iClusion (Dutch) clinical-trial database (Dec. 2020) from iClusion (Rotterdam, the Netherlands). The databases were aggregated and harmonized using the HMF knowledgebase-importer (v1.7). This list was manually corrected for discrepancies and subsequently, we curated the linked putative treatments for current on- and off-label aNEN and aNEN-subtype treatment options, as defined within the Netherlands by the Dutch Medicines Evaluation Board ("College ter Beoordeling van Geneesmiddelen; CBG)[100].

**Reporting summary**. Further information on research design is available in the Nature Research Reporting Summary linked to this article.

## Data availability

The WGS and corresponding clinical data used in this study was made available by the Hartwig Medical Foundation (Dutch nonprofit biobank organization) after signing a license agreement stating data cannot be made publicly available via third-party organizations. Therefore, the data are available under restricted access and can be requested upon by contacting the Hartwig Medical Foundation (https://www.hartwigmedicalfoundation.nl/applying-for-data/) under the accession code DR-036[27]. Within this manuscript, we furthermore made use of the actionable gene-variant and associated drug databases of CiViC (01-Nov-2018; https://civicdb.org/downloads/01-Nov-2018/01-Nov-2018-Clinical EvidenceSummaries.tsv), OncoKB (Nov. 2018; https://www.oncokb.org/actionableGenes), CGI (Nov. 2018; https://www.cancergenomeinterpreter.org/biomarkers) and the iClusion (Dutch) clinical trial database (Dec. 2020) from iClusion (Rotterdam, the Netherlands; Suppl. Data 1). The remaining data are available within the Article, Supplementary Information or available from the authors upon request.

## Code availability

Next to the initial processing workflows and software which are available at https://github.com/hartwigmedical/, any additional custom code and scripts used within this study (processing, analysis, and visualization) have been deposited on Bitbucket under the GPL-3.0 License: https://bitbucket.org/ccbc/dr-036_anen/.

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

## Acknowledgements

We would like to thank J. (Alberto) G. Nakauma Gonzalez for his assistance in implementing and describing the TAF calculations and iClusion for sharing their data on the association of genetic aberrations to actionable targets and clinical trials. Furthermore, we would like to thank the nationwide network and registry of histo- and cytopathology in the Netherlands (PALGA) for their assistance on retrieving and storing the pathological records of the included aNEN patients[79]. This publication and the underlying study have been made possible partly on the basis of the data that Hartwig Medical Foundation and the Center of Personalized Cancer Treatment (CPCT) have made available to the study.

## Author contributions

J.V.R., B.M., and H.J.G.V.D.W. wrote the manuscript, which all authors critically reviewed. J.V.R. and H.J.G.V.D.W. performed the bioinformatics analyses. B.M. managed clinical data assessment. F.A.L.M.E., M.T., L.M.V., H.K., M.W.D., and G.D.V. are clinical contributors. M.P.L. is PI of the CPCT-02 study, S.S. is chair of the CPCT and both supervise the CPCT-02 study. E.C. coordinated the sequencing of samples and contributed to the bioinformatics analyses.

## Competing interests

The authors declare no competing interests.
