## [Peer Review File · Nature Communications]

REVIEWER COMMENTS

Reviewer #1, expert in Precision medicine/statistics (Remarks to the Author):

This study performed WGS of biopsy specimens of 86 metastatic NE neoplasms including 16 NECs. These tumors are rare and these WGS data is very valuable. However, several reports are already published about genomic alterations or WGS of NE neoplasms derived from several organs, and the results from WGS and interpretations about these data are a little lack of novelty. In this manuscript, they should focus on novel findings and describe them, and some of main figures and descriptions are duplicated.

Overall, they should describe more statistical parameter or results in there analysis and it is required to analyze these valuable data by new concepts or analysis methods. Genomic analysis from WGS is not sufficient.

1) They should show the detail pathological information of all samples based on WHO criteria (grade, cell proliferation, cell division). The treatment information before biopsy are also important.

2) It is not clearly described about how different of genomic alteration or pattern between primary NENs and metastatic ones. Obviously NEC is quite different in many points, and pet and small-intestine NET are also different. They should describe more clearly about these issues.

3) The men (n = 16) reveals diploid to triploid genomes and a median TMB of 5.45 somatic mutations per Mb, which is in the midrange of TMB known for human primary cancers. Why did men have more TMB than met overall? Is there any men-specific and general mutational signature that can contributed to high TMB?

5) They say one sample show HRD and RAD51C germline variant, but it is not enough to have evidence of HRD.

6) Regarding to chromothripsis events, they detected chromothripsis-"like" events by Shatterseek. Is it enough to define true chromothripsis?

Why did some of men show chromothripsis or chromothripsis-like?

They also describe recurrent involvement of chr12 and extrachromosomal DNA of MDM2 or CCND2. This is a just speculation and they need more evidences for this interpretation.

7) Regarding to men of unknown primary localization, is it possible that they were primary tumor? Do they have any other histological components? Some studies suggested a possibility of trans-differentiation from adenocarcinoma to NEC in lung, prostate, and GI. These issue is very important in analysis of NEC and they should also address to this question by analyzing WGS data.

8) One met was strongly characterized by SBS36, associated with base excision repair (BER) deficiency due to MUTYH alterations, and they claimed a heterozygous germline pathogenic missense mutation within MUTYH (c.527A>G / p.Tyr176Cys; rs34612342). It is not significant because it happened in only one sample and this missense variant of MUTYH is wired. Do they have more evidence of pathogenicity of this variant?

9) Within men, an enrichment of alterations within TP53 (88% of men), KRAS (50%), RB1 (50%), MYC (31%), APC (31%), ZFH4 (31%), UBR5 (25%) and presence of kataegis (31%) could be appreciated ($q \leq 0.05$). In pancreas-derived met, an enrichment of was seen for MEN1 (40% of pancreas-derived met), ATRX (25%), DAXX (25%), SETD2 (25%) and PCNT (20%) whilst midgut-derived met revealed enrichment of CDKN1B alterations (25% of midgut-derived met). These driver genes for NET are not new. Any difference between met and primary NET which were reported by several studies.

What is new driver genes of men?

10) Mutations of TP53 are RB1 are drivers for NEC. Did they detect SVs of TP53 or RB1 by WGS? APC (31%) were frequently mutated in men? They overlapped colorectal cancer or intestinal cells?

KRAS is also frequently mutated in pancreatic cancer and is it possible that KRAS-mutated men are originated from pancreatic ductal cells?

11) Clinically-actionable mutations. They just annotated actionability from genomic alterations. Do they have any other evidences of actionability? For examples, some of mNEN patients in this cohort actually showed some response to these drugs?

Hjgh TMB is not enough to expect the response of ICB in other types of tumor than lung cancer, melanoma, and MSI+ CRCs.

Reviewer #2, expert in neuroendocrine neoplasms (Remarks to the Author):

The authors report on whole-genome sequencing of 86 metastatic neuroendocrine neoplasms. The strengths of the study is the strong methodology regarding sequencing, the detailed report of the bioinformatics methods. The main novelty regarding collective is the use of metastatic samples, we still have very limited knowledge on differences between primary tumors and metastases as well as on potential heterogeneity between metastases in neuroendocrine neoplasms.

There are several major weaknesses of the study: The mixture of many different entities, including neuroendocrine carcinomas of different organs as well as neuroendocrine tumors of different organs, leads to very small numbers of each tumor type in the end, limiting the room for significant novelties. Without matching primary tumors and metastases not much can be added on the understanding of metastases from a genomic standpoint. As a consequence, the authors report on the metastatic samples mainly genomic alterations that have been described before on primary tumors.

The section on potential therapeutic targets needs probably a more critical discussion, as there are several reports published reporting a lack of correlation of mutations of the mTor pathway to everolimus treatment for example, such statements should be included.

Specific:

Introduction, line 52: This separation is not merely a clinical separation, however this separation between neuroendocrine tumors and neuroendocrine carcinomas has been introduced by the WHO classification in 2000 and has been defined in more detail in 2017 and 2018 classifications. It is well accepted that these entities are genetically non-related, which is confirmed by the present analysis.

Introduction, line 64: The proposed statement is not entirely true for pancreatic neuroendocrine tumors, where DAXX/ATRX mutations have repeatedly been shown as of prognostic value (please cite). The important part of the statement however is true, that there is no predictive marker available. (Unfortunately, the collective as presented, is not suitable for defining a predictive marker either).

Introduction, line 70: While it is correct that whole-genome sequencing data is very limited, the availability of whole exom sequencing date and large panel sequencing date should be added here, especially as there are no large additional findings reported here using whole-genome sequencing. The main differences would be expected in non-coding regions, which, however, is difficult to analyse.

There are some lung neuroendocrine carcinomas included: if these tumors are to be retained in the dataset, a discussion and presentation of the available large published genomic data of lung neuroendocrine carcinomas should reported (including publications in Journals such as Nature).

Introduction, line 87: There are additional publications which the authors seem to have missed, for example on colonic neuroendocrine carcinomas and mixed adenoneuroendocrine carcinomas, for pancreatic neuroendocrine carcinomas and for lung neuroendocrine carcinomas. A comparison of the presented results to this sequencing projects should be added.

The results part is very well written and easy to follow for the genomic parts. A paragraph on clinical and pathological aspects is missing, this might be defined in the study protocol. It would be important to know about pretreatments and potential correlations (to escape mutations?) for example.

Page 7, line 211: Association with smoking is very clear in lung NEC, however it is expected in analogy to non-neuroendocrine carcinomas to be present in bladder NEC, esophageal NEC.

Line 273: ZFH4 and UBR5 appear more novel and could be further explained in the discussion part.

Line 275: The same seems true for PCNT in well-differentiated NET.

Line 286/287: The low mutation rate in Midgut-NET is very well known already. More important would potentially be the missing identification of potential drivers in non-coding regions of the genome. Detection of such alterations would be a big strengths of the methods applied.

Discussion, line 365: Citation and discussion of the American Study sequencing metastatic pancreatic neuroendocrine tumors after temozolomide treatment could be added.

Line 400-402: An explanation and expansion on the newly described genes could be added here.

Line 425-426: Here, the very low TMB of all NET could be stressed, the statement "increasing TMB" implies some high TMB, which is very exceptionally the case. Even the TMB of 5 in neuroendocrine carcinomas is lower than expected in analogy to lung NET, this observation could be emphasized.

Discussion Line 434-448: As the mTOR pathway mutations are known for many years, people have already looked for association with mTOR pathway mutations and response to everolimus, which was never published with a positive correlation. Such studies should be cited and discussed here. The 49 % of patients with "specific genomic alteration or genotype for which an FDA-approved drug is available" suggest probably a more pronounced role of these mutations as the present evidence allows.

Online methods: Who did evaluate the tumor content? No person with pathology training or no biobank is among the authors.

In line with this, how was the pathological diagnosis obtained? Was always as second biopsy core performed for this? Or was the pathology diagnosis based on earlier biopsies/resections specimens?

How was Ki67 index measured? From a parallel biopsy or from biopsies of different locations? The authors should indicate on their strategy towards heterogeneity.

Figure 1 second row: Where does the number 114X for whole-genome sequencing stem from? The same is true for the white box in the third row, repeating 114X.

Figure 1: Instead of foregut, "stomach" would probably be more precise if the figure is interpreted in the correct way. Pancreas and lung also belong to the foregut, and this separation of NET into foregut, midgut and hindgut is not recommended anymore.

Reviewer #3, expert in mutational analysis/genetics (Remarks to the Author):

In this study the authors have reported the mutational landscape of 86 whole-genome sequenced metastatic neuroendocrine neoplasms (mNEN). The main finding of their analysis was the delineation of distinct genomic subpopulations of mNEN based on primary localization and differentiation grade, with the mNEC derived from poorly differentiated neuroendocrine carcinomas

(NEC) and the mNET populations derived from the better differentiated neuroendocrine tumors (NET). These subpopulations were different in terms of tumor mutational burden, genomic stability, and distinct mutated driver genes. Furthermore, distinct drivers were enriched with somatic aberrations in pancreatic and midgut-derived neuroendocrine tumors. Finally, whole genome sequencing of metastatic lesions revealed 49% of the analyzed mNEN patients harbored clinically-relevant targetable somatic aberrations indicating a potential extension of the current treatment options.

To my view, the paper is a potential good paper especially if considering the novelty of the aim; besides, the methodology used for data analysis and the results are very interesting and, if extended to additional samples, could become clinically relevant. The scientific content seems good and the English style and language used in the manuscript are also good even though the presence of some typing errors would require a recheck of the text. Moreover, I recognize that the background is accurately written even though it can be further improved. The methods performed to analyze the whole genome sequencing data seem adequate, technically sound and properly employed. Indeed, most of them were also applied in other previously published and high-quality studies. Overall, the findings are interesting and quite well-organized in each section of the Results. Moreover, the results were clearly described and data analyses were interpreted in a comprehensible manner. Besides, the figures seem of the right quality for the journal. No remarkable incongruences could be observed throughout the text. The paper provides sufficient evidence to support the conclusions stated in the discussion section.

Strengths: It is a noteworthy research topic with high novelty value. Indeed, this study is the first to have investigated the whole genome and mutations of a cohort of 86 metastatic NEN from various primary localizations and differentiation grade. Previous studies have analyzed only few specimens of metastasis from NEN cases. Thus, it may represent an advance in the field and is likely to be a pioneer study in the categorizing these tumors.

Weakness/Limitations: Given the rarity of this disease, a relatively limited number of NEN cases with whole genome sequencing data have been analyzed in this study. Thus, a basic concern would be whether the sample has statistical value. A further weakness is represented by the lack of correlation between the mutation signatures and patient outcomes.

Specific comments:

Introduction

1. Example of language and typing errors:

Line 51: Neuroendocrine neoplasms (NEN) is a heterogeneous.....; is should be changed with are. Accordingly, the whole text should be rechecked for similar errors.

2. References in the text:

Lines 51-68: Only one reference is quoted. Are all the sentences from the same reference #1? In my opinion, some sentences need suitable bibliographic citations.

Results

3. To avoid confusion for the reader in the initial part of results' description some modifications should be applied. To my point of view, characteristics of patients should be described all together and panels C and D of Supp Figure 2 should be included in Figure 1. Lines 115-120 should be moved above (line 108).

Sequencing characteristics can remain in the supp Figure 2 but its description should be moved at the beginning of the second paragraph of the Results ("The mutational landscape").

4. Line 194: SBS should be quoted at line 175.

5. Line 196: comma should be moved after the parenthesis.

6. Line 233: Currently, several methods and algorithms have been developed to identify driver genes in tumor exomes. The authors should briefly justify the reasons of their choices (GISTIC2 and dN/dS) for this kind of analysis.

Discussion

7. I understand that previous studies have analyzed only few specimens from mNEN cases. However, the discussion of the obtained results should consider also a comparison with previous publications, clearly outlining similar and/or distinct results as well as comparing the utilized

approaches.

8. Likewise, there are several publicly available whole-genome sequencing data from many types of tumor samples that would offer the opportunity to integrate the findings of this study with comparable analyses on independent cohorts (if any). The authors should discuss this point also as a future perspective.

Reviewer #1 - Expert in Precision medicine/statistics

Comment #1 - Incorporation of pathological and clinical information

They should show the detail pathological information of all samples based on WHO criteria (grade, cell proliferation, cell division). The treatment information before biopsy is also important.

Response:

No uniform pathological reports have been made of the fresh-frozen biopsies of the metastatic samples since these biopsies were harvested only for the CPCT-02 study. However, we have added the pathological overview of the available diagnostic tumor samples into the manuscript using the nation-wide (Dutch) PALGA system. We stress that these are not the metastatic tissues on which WGS was performed, but rather the respective metastatic tissues at diagnosis or, if the metastatic tumor was not available, any primary lesion (**line 528 - 539**). We've incorporated this information (grade and proliferation index) as tracks below our landscape figures to provide an overview and to put this into context our sequencing / molecular data (**Fig. 2-4** and **Suppl. Fig. 3**)

We furthermore added the pre-treatment history of the patients to the manuscript (**Suppl. Fig. 1d**). Given the heterogeneity of the patient population and given treatments, we argue that any general conclusion regarding based on their analysis will fall short due to the limited number of patients per (generalized) treatment group and therefore do not pursue this further.

Whilst reviewing the pathological records, we encountered the possibility that we previously included a very rare form of malignancy, namely a gastrointestinal neuroectodermal tumor (GNET). After discussing with fellow co-authors and colleagues within the field, we decided to exclude this patient from our study. This made us retain only **85** mNEN, rather than the originally included 86 samples. As a result, we had to fully re-perform all analyses, rewrite the manuscript and remake all figures to reflect this change of total sample size. This did not significantly affect major results but did slightly alter median, IQR values and genetic aberrations close to statistical significance throughout the manuscript.

Comment #2 - Comparison to primary NEN

It is not clearly described about how different of genomic alteration or pattern between primary NENs and metastatic ones. Obviously, NEC is quite different in many points, and pet and small-intestine NET are also different. They should describe more clearly about these issues.

Response:

We've extended our literature search and thoroughly reviewed existing peer-reviewed scientific literature regarding somatic aberrations within primary NEN. We argue that further, more in-depth comparisons (e.g., statistical analysis upon the mutational frequencies or (non-coding) aberrations), between primary and metastatic setting are of very limited additional value within this manuscript as this carries the risk of over- and misinterpretation, as these cohorts, the patients and their clinical history vary widely within their captured NEN-population and sequencing/molecular techniques. Furthermore, such an undertaking risks spiraling into a review-like effort whilst comparing the heterogeneous populations only on a superficial level; as the underlying data is often not publicly-available or requires a uniform re-analysis prior to interpretation.

A paired comparison between matched primary and metastatic lesions from the same patients would be the best experimental setup to delve more deeply within this interesting field of research, however these cases are not included within our current metastatic cohort.

Comment #3 - Mechanisms driving increased TMB in mNEC vs. mNET.

The men ($n = 16$) reveal diploid to triploid genomes and a median TMB of 5.45 somatic mutations per Mb, which is in the midrange of TMB known for human primary cancers. Why did men have more TMB than met overall? Is there any men-specific and general mutational signature that can contribute to high TMB?

Response:

We performed an additional investigation into identifying the mechanisms capable of significantly increasing the TMB within mNEC compared to mNET. We extended our mutational signature analyses to the latest COSMIC (v3.1) mutational signature database which includes single base substitutions (SBS), InDels and doublet-base signatures and performed restrictive mutational signature refitting using the MutationalPatterns (v1.3.0) package on all mNEC ($n = 16$) and mNET ($n = 69$) samples yet this did not reveal additional patterns not seen in the prior mutational signature analysis (COSMIC v3 - SBS-only and *de novo* NMF).

We did however extend our analysis by investigating differences in the relative contribution of the previously-established COSMIC (v3) mutational signatures between our major subgroups (mNEC, midgut- and pancreas-derived mNET) as incorporated within **Suppl. Fig 6g** and further described in **line 232 - 239**. No immediate evidence suggesting a distinct mechanism could be found associated to the high(er) mutational frequency within mNEC compared to mNET.

Comment #4 - Lack of evidence that *RAD51C* drives HRD in mNEN.

They say one sample shows HRD and *RAD51C* germline variant, but it is not enough to have evidence of HRD.

Response:

We have considered this comment and found that we perhaps did not document and introduce our chosen HRD-detection approach extensively enough. We have extended the explanation of how we determine HRD within samples in the methods section (**line 190 - 194 & 656 - 665**).

Briefly, we make use of CHORD¹; a random-forest based classifier designed to classify samples with evidence of HRD (*BRCA1*-type, *BRCA2*-type or otherwise) by using all the information captured within all the somatic small mutations and somatic structural variants of whole-genome sequenced samples. If a sample contains sufficient HRD-related genomic scars (structural variants) and additional markers for HRD, that sample will be classified as HRD.

We next associated the single HR-deficient sample within our cohort with a somatic pathogenic *RAD51C* mutation which in turn facilitated the multitude of genomic scars seen within this sample (upon which the classification was based through CHORD). The role of *RAD51C* in DNA damage repair and, if mutated by mono- or bi-allelic mutations, in genomic scarring/HRD has been confirmed by other scientific groups.¹⁻⁴

Comment #5 - Chromothripsis

- 1) Regarding to chromothripsis events, they detected chromothripsis-“like” events by Shatterseek. Is it enough to define true chromothripsis?

- 2) Why did some of men (mNEC) show chromothripsis or chromothripsis-like?
- 3) They also describe recurrent involvement of chr12 and extrachromosomal DNA of *MDM2* or *CCND2*. This is a just speculation and they need more evidences for this interpretation.

Response:

As detailed by Korbel and Campbell⁵ and extended by Cortés-Ciriano et al.⁶, the occurrence of chromothripsis has been described by the following set of (connected) criteria:

- Clusters of random interleaved structural variants with equal distributions of the type of structural variants (inversions, deletions and tandem duplications).
- An enrichment of structural variants within chromosomes.
- Oscillating copy-number segments within the chromothripsis region.

Shatterseek aims to capture this information and to provide a computational method to determine chromothripsis within whole-genome sequences samples. We maintained the criteria as detailed by Cortés-Ciriano et al. to determine 'true' chromothripsis events. We furthermore visually inspected the chromothripsis events of all six chromothripsis-occurring samples (**Suppl. Fig. 5**) and observed these criteria to indeed be in place.

We could not find any leads as to why these samples (both mNEC and mNEN) harbored chromothripsis compared to the other samples. As postulated by Cortés-Ciriano et al., the (indirect) role of chromothripsis could be to drive aberrant expression of driver genes (through amplification of oncogenes, deletion of tumor suppressor genes, promoter hijacking or otherwise). However, we have too few mNEN samples to further investigate this in more detail, yet we made the possible association to *MDM2* or *CCND2* involvement. In hindsight, we agree that we do not hold enough evidence to substantiate this claim and have removed this association from the manuscript.

Comment #6 - Trans-differentiation and primary disease

Regarding to men (mNEN) of unknown primary localization, is it possible that they were primary tumor? Do they have any other histological components?

Some studies suggested a possibility of trans-differentiation from adenocarcinoma to NEC in lung, prostate, and GI. This issue is very important in the analysis of NEC and they should also address this question by analyzing WGS data.

Response:

An interesting observation and remark as treatment-emergent trans-differentiation is seen more often as a remarkable route to achieve treatment-resistance. However, as trans-differentiation is driven mostly by epigenetic dysregulation, few changes within the DNA are observed. We can however make use of the history of the somatic cell to deduce those of likely non-neuroendocrine origin, such as the *TMPRSS2-ERG* fusion in treatment-emergent neuroendocrine prostate adenocarcinoma (t-PNET). However, no samples contained genomic evidence suggesting non-neuroendocrine origins and we are further confident by the absence of admixed (adenoma)carcinoma components within the pathology reports of the primary disease.

Nevertheless, for some patients the tissue of the primary tumor wasn't available and the diagnosis was made on metastatic tissue alone. For these patient's lacking primary tissue pathology, we cannot fully exclude that the primary tumor showed signs of [a component of] adenocarcinoma beyond lacking striking genomic features related to other origins.

Comment #7 - Germline *MUTYH* aberration driving SBS36

One met (mNET) was strongly characterized by SBS36, associated with base excision repair (BER) deficiency due to *MUTYH* alterations, and they claimed a heterozygous germline pathogenic missense mutation within *MUTYH* (c.527A>G / p.Tyr176Cys; rs34612342).

It is not significant because it happened in only one sample and this missense variant of *MUTYH* is wired. Do they have more evidence of pathogenicity of this variant?

Response:

We do agree that this is not a recurring feature within our cohort and all interpretation has been performed on only a single case, yet we strongly believe that the stated *MUTYH* mutation could indeed generate the *MUTYH*-like genomic scarring in combination with the observed *MUTYH* mutational signature (G:C > T:A base excision repair) within this sample. This behavior has previously been reported within neuro-endocrine cells within external studies.^{7,8}

Furthermore, this particular (germline) mutation (rs34612342(G)) has previous been linked to *MUTYH*-associated polyposis and hereditary cancer predisposition (<https://www.ncbi.nlm.nih.gov/clinvar/variation/5293/>). We therefore believe strongly that the stated association between this germline *MUTYH* aberration and the observed signature can remain as-is due to previous reports of this phenomenon in context with this particular germline *MUTYH* aberration and the somatic loss of as single chromosome 1.

Comment #8 - Novel driver genes of mNEN.

Within men, an enrichment of alterations within *TP53* (88% of men), *KRAS* (50%), *RB1* (50%), *MYC* (31%), *APC* (31%), *ZFH4* (31%), *UBR5* (25%) and presence of kataegis (31%) could be appreciated ($q \leq 0.05$). In pancreas-derived met, an enrichment of was seen for *MEN1* (40% of pancreas-derived met), *ATRX* (25%), *DAXX* (25%), *SETD2* (25%) and *PCNT* (20%) whilst midgut-derived met revealed enrichment of *CDKN1B* alterations (25% of midgut-derived met).

These driver genes for NET are not new. Any difference between mNET and primary NET which were reported by several studies. What are the new driver genes of mNEN?

Response:

We optimized our analysis (on $n = 85$ samples) to detect mutually-exclusive gene-aberrations (**line 297 - 305 & 666 - 677**). We re-designed our test to only determine the enrichment of mutant genes within one of our major subgroups (mNEC, mNET - Midgut and mNET - Pancreas) using a Fisher's Exact test. In addition, we narrowed our selection of genes to test within this analysis using the following scheme; 1) genes identified within focal peaks by GISTIC2, 2) genes identified by dN/dS, 3) genes containing coding mutations in $\geq 20\%$ of samples within each respective major subgroup, and 4) genes containing deep deletions or deep amplifications in $\geq 20\%$ of samples within each respective major subgroup (**Figure 3, 4 and Sup. Fig 8e**).

We've furthermore extended the discussion (**line 431 - 462**) to discuss these findings and to focus more on the novel (or less-known) detected aberrations from our dN/dS, GISTIC2 and mutually-exclusive analysis including the observation of *CSMD1* and *CSMD3* enriched in mNEC.

Comment #9 - Overlap of SV and pancreatic *KRAS* mutations.

Mutations of *TP53* are *RB1* are drivers for NEC. Did they detect SVs of *TP53* or *RB1* by WGS?

APC (31%) were frequently mutated in men? They overlapped colorectal cancer or intestinal cells? *KRAS* is also frequently mutated in pancreatic cancer and is it possible that *KRAS*-mutated men are originated from pancreatic ductal cells?

Response:

As part of our present analysis, we determined the overlap of structural variants within segments of genes (i.e., not all exons) and included these as “Structural Variants” within **Fig. 3 and 4**. A single sample (mNEC) showed such a structural variant within *RB1*. In addition, we make use of the LINX / PURPLE / GRIDDS⁹ suite which detects structural variants from WGS and incorporates this information in determining the somatic copy-number segments. Hence, structural variants were also included in determining copy-number alterations as shown in **Fig. 3, 4 and supplementary table 1**.

The reviewer does indeed allude to the promise of using WGS to further detail the potential origin and mechanisms behind the formation of (recurrent) structural variants, e.g., loss of *RB1* or *TP53*. Using the full catalogue of the CPCT-02 and DRUP-obtained whole-genome sequenced metastatic samples, the next steps can be undertaken into furthering our understanding behind such mechanisms and these efforts are currently underway in several other projects. Unfortunately, the presented mNEN cohort of 85 samples does not readily reveal such (recurrent) patterns for proper interpretation, this is likely due to the overall low mutational burden and, a for these types of analysis, low sample size. We did not perform additional comparisons against colorectal or intestinal cells as we deemed this beyond the scope of this project and are confident that we selected samples of primary neuroendocrine origin (as further evidenced by the obtained pathological information). In addition, we detect *KRAS* aberrations mostly in our mNEC population (50% of mNEC vs. 5.1 in mNET) which follows previous reported incidences in primary NEC.^{10–12}

Comment #10 - Clinically-actionable mutations.

They just annotated actionability from genomic alterations. Do they have any other evidences of actionability? For examples, some of mNEN patients in this cohort actually showed some response to these drugs?

High TMB is not enough to expect the response of ICB in other types of tumor than lung cancer, melanoma, and MSI+ CRCs.

Response:

Although we do not have clinical data supporting the benefit of these proposed therapies within the presented mNEN patients, we feel that this is an important addition to this manuscript as it could spark the investigation of new treatment avenues within mNEN.

Regarding the chances of benefit from immune-checkpoint inhibitors, we are indeed gaining more knowledge for more malignancies deriving benefit from this treatment; especially when patient selection is performed based on TMB. To support this, it has been reported that 2 out of 5 neuroendocrine tumors which harbored high TMB (as defined using the same definition used in our manuscript), showed an objective response to pembrolizumab.¹³ Therefore, we believe that the observation and reporting of high TMB in mNEN does open the door to a potential new treatment strategy.

We have however made it clearer that these associations are yet-to-be tested and merely reveal the potential landscape of current and experimental therapies. We have extended the discussion

Response to reviewers concerning: In-depth analysis of the genomic landscape of 85 metastatic neuroendocrine neoplasms reveals subtype-heterogeneity and potential therapeutic targets by van Riet et al.

by adding this information and addressing previous reports with mixed success from these experimental therapies in mNEN (**line 482 - 488**).

Reviewer #2 - Expert in neuroendocrine neoplasms

The authors report on whole-genome sequencing of 86 metastatic neuroendocrine neoplasms. The strengths of the study are the strong methodology regarding sequencing, the detailed report of the bioinformatics methods. The main novelty regarding collective is the use of metastatic samples, we still have very limited knowledge on differences between primary tumors and metastases as well as on potential heterogeneity between metastases in neuroendocrine neoplasms.

There are several major weaknesses of the study:

- 1) The mixture of many different entities, including neuroendocrine carcinomas of different organs as well as neuroendocrine tumors of different organs, leads to very small numbers of each tumor type in the end, limiting the room for significant novelties.
- 2) Without matching primary tumors and metastases not much can be added on the understanding of metastases from a genomic standpoint. As a consequence, the authors report on the metastatic samples mainly genomic alterations that have been described before on primary tumors.
- 3) The section on potential therapeutic targets needs probably a more critical discussion, as there are several reports published reporting a lack of correlation of mutations of the mTor pathway to everolimus treatment for example, such statements should be included.

Response:

The mixture of different entities indeed resulted in smaller major subgroups; however, we are confident that the presented cohort and overview captures the most commonly encountered entities of mNEN and provides a comprehensive list of common genetic drivers of mNEN. The relatively small numbers of captured samples reflect the rare occurrence of these entities as seen in daily clinical practice with the benefit of whole-genome sequencing however enabling us to fully capture the individual genomic aspects. We aimed to first focus on and report on the catalogue of coding aberrations within the mNEN cohort to allow for a more focused interpretation of common genetic drivers.

We fully agree that the subsequent efforts in unraveling the complex mechanisms underlying the progression of primary disease towards metastatic disease need to be investigated using paired primary and metastatic samples of the same patient. This was unfortunately not part of the primary goals of the CPCT-02 and DRUP studies (of which this cohort is a subset). Please also see comment #2 of reviewer #1 for further detail on this effort.

Additionally, we have carefully re-inspected the discussion on the potential therapeutic efforts and down-toned our conclusions.

Comments #1 - Specific comments concerning Introduction

- 1) *Separation of NET/NEC according to WHO*
 - Introduction, line 52: This separation is not merely a clinical separation, however this separation between neuroendocrine tumors and neuroendocrine carcinomas has been introduced by the WHO classification in 2000 and has been defined in more detail in 2017 and 2018 classifications. It is well accepted that these entities are genetically non-related, which is confirmed by the present analysis.
- 2) *DAXX/ATRX mutation for pNET.*
 - Introduction, line 64: The proposed statement is not entirely true for pancreatic neuroendocrine tumors, where DAXX/ATRX mutations have repeatedly been shown as of prognostic value (please cite). The important part of the statement

however is true, that there is no predictive marker available. (Unfortunately, the collective as presented, is not suitable for defining a predictive marker either).

- 3) *Make mention of current WES and panel-based sequencing datasets.*
 - Introduction, line 70: While it is correct that whole-genome sequencing data is very limited, the availability of whole exon sequencing data and large panel sequencing data should be added here, especially as there are no large additional findings reported here using whole-genome sequencing. The main differences would be expected in non-coding regions, which, however, is difficult to analyze.
 - There are some lung neuroendocrine carcinomas included: if these tumors are to be retained in the dataset, a discussion and presentation of the available large published genomic data of lung neuroendocrine carcinomas should be reported (including publications in Journals such as Nature).
- 4) *Re-evaluate sequencing efforts / publications of NEN.*
 - Introduction, line 87: There are additional publications which the authors seem to have missed, for example on colonic neuroendocrine carcinomas and mixed adenoneuroendocrine carcinomas, for pancreatic neuroendocrine carcinomas and for lung neuroendocrine carcinomas. A comparison of the presented results to this sequencing project should be added.

Response:

- 1) We have extended the introduction with additional references to the IARC/WHO criteria concerning NET/NEC for clarification. **(line 52 - 58)**
- 2) We have corrected and addressed this by rephrasing and referencing the suggested studies. **(line 64 - 66)**
- 3) As part of the extension of the manuscript detailing the major finding within primary NEN and their occurrence with mNEN, we have expanded the introduction with associations to previous sequencing efforts of NEN **(line 73 - 90)**.

Beyond the in-depth catalogue of structural variations and mutational signatures made possible due to WGS within our cohort, we have also further highlighted the currently still-unexplored aspects of the non-coding regions. However, due to the limited number of samples and the current complexity of interpreting these findings, this can hopefully be unraveled by future efforts and increased knowledge on how to accurately exploit and analyze these regions **(line 506 - 513)**.

- 4) We have now also included these sequencing efforts, please see previous responses on extending the prior sequencing efforts.

Comments #2 - Specific comments concerning Results

- 5) *Incorporation of pathological and clinical information.*
 - The results part is very well written and easy to follow for the genomic parts.
 - A paragraph on clinical and pathological aspects is missing, this might be defined in the study protocol. It would be important to know about pretreatments and potential correlations (to escape mutations?) for example.
- 6) *Association of smoking.*
 - Page 7, line 211: Association with smoking is very clear in lung NEC, however it is expected in analogy to non-neuroendocrine carcinomas to be present in bladder NEC and esophageal NEC.
- 7) *Increased mention of novel drivers.*
 - Line 273: *ZFH4* and *UBR5* appear more novel and could be further explained in the discussion part.
 - Line 275: The same seems true for *PCNT* in well-differentiated NET.
- 8) *Detection of non-coding aberrations in midgut-derived mNET.*
 - Line 286/287: The low mutation rate in Midgut-NET is very well known already. More important would potentially be the missing identification of potential

drivers in non-coding regions of the genome. Detection of such alterations would be a big strength of the methods applied.

Response:

- 5) We have added the pathological information (for the most-recently available primary or metastatic tumor sample) and the pre-treatment information to the manuscript (**Suppl. Table 1; Suppl. Fig 1d and as annotation track within figures; line 528 - 539**). Yet given the heterogeneity of the patient population and said given treatments, we argue that any analysis will fall short due to the limited number of patients per (generalized) treatment group.
- 6) Interestingly, the mutational signature related to smoking (and incidence) is different between lung malignancies (be they NEC or other forms of pulmonary carcinoma) and bladder malignancies. In our cohort, no NEC known to have arisen in the urinary bladder are included. A single mNEC from the esophagus has been included in our cohort, however this sample does not harbor any of the known mutational signatures (COSMIC v3) associated with smoking.
- 7) Based on the reviewer comments, we have optimized our strategies in detecting potentially novel drivers and have placed more emphasis onto our novel genes compared to previously known NEN drivers within the manuscript (for further details, please see **comment #8 of reviewer #1**).
- 8) Please also see the introduction to reviewer #2. We wholeheartedly agree that exploring the non-coding regions of this malignancy still holds many unexplored areas of research and potential avenues for understanding the enigmatic nature of these low-TMB malignancies. However, we feel that such an undertaking would be better appreciated in an even larger mNEN cohort, as our preliminary non-coding analyses (data not shown in manuscript) showed only very limited number of overlapping loci which complicates proper interpretation. Furthermore, we feel that the field of bioinformatics which focuses on the non-coding genome is not yet as well-developed as the field of the analysis and interpretation of the coding portion of the genome. As mentioned, we therefore feel it better to relegate the further investigation of the non-coding genome of mNEN into a future (perhaps multi-institutional) effort and focus on the more robust analysis of the presented work in this manuscript.

Comments #3 - Specific comments concerning Discussion

- 9) *Add missing discussion of pNET after temozolomide treatment.*
 - Discussion, line 365: Citation and discussion of the American Study sequencing metastatic pancreatic neuroendocrine tumors after temozolomide treatment could be added.
- 10) *Association of smoking.*
 - Page 7, line 211: Association with smoking is very clear in lung NEC, however it is expected in analogy to non-neuroendocrine carcinomas to be present in bladder NEC and esophageal NEC.
- 11) *Description of novel genes.*
 - Line 400-402: An explanation and expansion on the newly described genes could be added here.
- 12) *Rewording of 'high TMB' in regards to mNEN*
 - Line 425-426: Here, the very low TMB of all NET could be stressed, the statement "increasing TMB" implies some high TMB, which is very exceptionally the case. Even the TMB of 5 in neuroendocrine carcinomas is lower than expected in analogy to lung NET, this observation could be emphasized.
- 13) *Re-evaluate clinically-actionable mutations.*
 - Discussion Line 434-448: As the mTOR pathway mutations are known for many years, people have already looked for association with mTOR pathway mutations and response to everolimus, which was never published with a positive correlation. Such studies should be stated and discussed here. The 49 % of patients with "specific genomic alteration or genotype for which an FDA-

approved drug is available” suggest probably a more pronounced role of these mutations as the present evidence allows.

Response:

- 9) We apologize for the omission and added this reference to our findings (**line 405 - 407**).
- 10) Please see the above response on **comment #6 of reviewer #2**.
- 11) We have indeed added more emphasis on our novel findings compared to previous report. Please also see **comment #8 of reviewer #1**.
- 12) We have indeed placed the overall strikingly low TMB observed in mNEN into better perspective to highlight that increase is only markedly within mNEN itself, with perhaps the exception of mNEC; which itself is *still* not surprisingly high compared to other cancers (**line 468 - 473**).
- 13) Please see **comment #10 of reviewer #1**. We have indeed down-toned our discussion and highlight that these results give promise to possible extensions of the treatment repertoire yet that these are preliminary observations without, in most cases, backing evidence for treatment efficiency in mNEN.

Comments #4 - Specific comments concerning Online Methods and Figures

- 14) *Elaborate on the method of tumor content estimation and pathological information.*
 - Online methods: Who did evaluate the tumor content? No person with pathology training or no biobank is among the authors.
 - In line with this, how was the pathological diagnosis obtained? Was always as second biopsy core performed for this? Or was the pathology diagnosis based on earlier biopsies/resections specimens?
 - How was Ki67 index measured? From a parallel biopsy or from biopsies of different locations? The authors should indicate on their strategy towards heterogeneity.
- 15) *Elaborate on the sequencing depth and re-evaluate ‘foregut’ description. (figure 1)*
 - Figure 1 second row: Where does the number 114X for whole-genome sequencing stem from? The same is true for the white box in the third row, repeating 114X.
 - Figure 1: Instead of foregut, “stomach” would probably be more precise if the figure is interpreted in the correct way. Pancreas and lung also belong to the foregut, and this separation of NET into foregut, midgut and hindgut is not recommended anymore.

Response:

14) The Hartwig Medical Foundation (HMF; i.e., the sequencing partner) employs an expert pathologist who determines the tumor content of each tissue slide prior to sequencing (min. 30% tumor percentage). In addition, a fragment of the constructed sequenced library from the tumor material is first sequenced to estimate the tumor percentage using *in silico* methods (as described by Priestley et al.) prior to full-scale sequencing.

No parallel biopsy was taken for these metastatic samples, all pathological information was obtained from a prior biopsy or resection specimen of the patient. The Ki67 index was also assessed on these samples obtained for diagnosis. Indeed, heterogeneity of Ki67 staining is an important issue in the treatment of mNEN. As we obtained the Ki7 index from real-world diagnostic samples, no further information on Ki67 staining in other lesions or other parts of the biopsied lesions is available.

15) We updated our ‘Foregut’ sample into its correct classification as ‘Gastric’. (**Suppl. Table 1 and Figure 1-3**). As it only constituted a single sample, this was indeed a better option. We’ve also added the explanation of the 114x (now 107x) and 38x within the legend of Figure 1 (**line 943 - 945**), these were the median avg. read coverage per base for the tumor and reference peripheral blood sample, respectively.

Reviewer #3 - Expert in mutational analysis/genetics

In this study the authors have reported the mutational landscape of 86 whole-genome sequenced metastatic neuroendocrine neoplasms (mNEN). The main finding of their analysis was the delineation of distinct genomic subpopulations of mNEN based on primary localization and differentiation grade, with the mNEC derived from poorly differentiated neuroendocrine carcinomas (NEC) and the mNET populations derived from the better differentiated neuroendocrine tumors (NET). These subpopulations were different in terms of tumor mutational burden, genomic stability, and distinct mutated driver genes.

Furthermore, distinct drivers were enriched with somatic aberrations in pancreatic and midgut-derived neuroendocrine tumors. Finally, whole genome sequencing of metastatic lesions revealed 49% of the analyzed mNEN patients harbored clinically-relevant targetable somatic aberrations indicating a potential extension of the current treatment options.

To my view, the paper is a potential good paper especially if considering the novelty of the aim; besides, the methodology used for data analysis and the results are very interesting and, if extended to additional samples, could become clinically relevant. The scientific content seems good and the English style and language used in the manuscript are also good even though the presence of some typing errors would require a recheck of the text. Moreover, I recognize that the background is accurately written even though it can be further improved. The methods performed to analyze the whole genome sequencing data seem adequate, technically sound and properly employed. Indeed, most of them were also applied in other previously published and high-quality studies. Overall, the findings are interesting and quite well-organized in each section of the Results. Moreover, the results were clearly described and data analyses were interpreted in a comprehensible manner. Besides, the figures seem of the right quality for the journal. No remarkable incongruences could be observed throughout the text. The paper provides sufficient evidence to support the conclusions stated in the discussion section.

Strengths: It is a noteworthy research topic with high novelty value. Indeed, this study is the first to have investigated the whole genome and mutations of a cohort of 86 metastatic NEN from various primary localizations and differentiation grade. Previous studies have analyzed only few specimens of metastasis from NEN cases. Thus, it may represent an advance in the field and is likely to be a pioneer study in the categorizing these tumors.

Weakness/Limitations: Given the rarity of this disease, a relatively limited number of NEN cases with whole genome sequencing data have been analyzed in this study. Thus, a basic concern would be whether the sample has statistical value. A further weakness is represented by the lack of correlation between the mutation signatures and patient outcomes.

Comments #1 - Specific comments concerning Introduction

1) Spelling and typo's

- *Example of language and typing errors: Line 51: Neuroendocrine neoplasms (NEN) is a heterogeneous.....; is should be changed with are. Accordingly, the whole text should be rechecked for similar errors.*

2) Re-evaluate references in the text

- Lines 51-68: Only one reference is quoted. Are all the sentences from the same reference #1? In my opinion, some sentences need suitable bibliographic citations.

Response:

- 1) We've corrected several regional and typing errors throughout the manuscript by carefully re-reading the whole manuscript and asking input from our English colleagues.
- 2) We have re-checked and extended the list of references used. In particular for quoted line 51 - 68, we indeed missed a reference to the IACR/WHO molecular/clinical classification scheme as also mentioned by comment #1 of reviewer #2.

Comments #2 - Specific comments concerning Results

- 3) *Reshuffling of related results.*
 - To avoid confusion for the reader in the initial part of results' description some modifications should be applied. To my point of view, characteristics of patients should be described all together and panels C and D of Supp Figure 2 should be included in Figure 1.
 - Lines 115-120 should be moved above (line 108).
 - Sequencing characteristics can remain in the supp Figure 2 but its description should be moved at the beginning of the second paragraph of the Results ("The mutational landscape").
- 4) *Line 194: SBS should be quoted at line 175.*
- 5) *Line 196: comma should be moved after the parenthesis*
- 6) *Line 233: Currently, several methods and algorithms have been developed to identify driver genes in tumor exomes. The authors should briefly justify the reasons of their choices (GISTIC2 and dN/dS) for this kind of analysis.*

Response:

- 3) We have indeed added our previous **Sup. Fig. S2C and S2D** into **Fig. 1** (as **Fig. 1c** and **1d**, respectively) to allow readers to more quickly assess the captured mNEN population. We've also shuffled the description of the mNEN cohort and sequencing information as suggested. Rather than placing the sequencing information into the Results section, we now first introduce the mNEN cohort (patient characteristics and generalized pre-treatment) and subsequently mention the sequencing protocol and depth. This indeed makes for a clearer description. In addition, we've also introduced previously-related NEN-drivers within the introduction and now refer to these studies in the later discussion for easier interpretation for readers.
- 4) SBS is now introduced at the first usage (**line 202**) within the results-section rather than only within the M&M. We apologize for the omission.
- 5) We apologize; however, we could not find any misplaced comma after a parenthesis within the given lines.
- 6) We have extended our reasoning as why to use the well-established dN/dS and GISTIC2 algorithms (**line 265 - 268**). Furthermore, we had success in discovering driver genes using these two applications before within our previous investigation of the whole-genome sequenced prostate and breast cancer cohorts of the CPCT-02 study.^{14,15} Hence, this is why we employed these applications again.

Comments #2 - Specific comments concerning Discussion

- 7) *Outlining of results against other NEN sequencing efforts*
 - I understand that previous studies have analyzed only few specimens from mNEN cases. However, the discussion of the obtained results should consider also a comparison with previous publications, clearly outlining similar and/or distinct results as well as comparing the utilized approaches.
- 8) *Extension of possible findings to additional tumor types to determine mNEN-specific alterations.*
 - Likewise, there are several publicly available whole-genome sequencing data from many types of tumor samples that would offer the opportunity to integrate the findings of this study with comparable analyses on independent

cohorts (if any). The authors should discuss this point also as a future perspective.

Response:

- 7) This valid point has also been raised by the other reviewers and we have therefore extended the introduction by highlighting the major prior findings and studies regarding the genomics of NEN and placing our results in perspective against previous results within NEN and mNEN (line 73 - 90 and 444 - 462).
- 8) We have extended the discussion by postulating several future efforts to deduce additional genomic alterations specific to mNEN and potentially driving mNEN. We opted to not go into detail on specific methodologies to perform said analyses but rather to spark interest within the scientific community to pursue and integrate these large-scale datasets as the exact methodologies will likely differ with the ever-increasing knowledge and algorithms to interrogate these data-sets. We do indeed share that this is an important concluding statement (line 506 - 513).

References

1. Nguyen, L., W. M. Martens, J., Van Hoeck, A. & Cuppen, E. Pan-cancer landscape of homologous recombination deficiency. *Nat. Commun.* (2020) doi:10.1038/s41467-020-19406-4.
2. Somyajit, K., Subramanya, S. & Nagaraju, G. RAD51C: A novel cancer susceptibility gene is linked to Fanconi anemia and breast cancer. *Carcinogenesis* (2010) doi:10.1093/carcin/bgq210.
3. Vaz, F. *et al.* Mutation of the RAD51C gene in a Fanconi anemia-like disorder. *Nat. Genet.* (2010) doi:10.1038/ng.570.
4. Min, A. *et al.* RAD51C-deficient cancer cells are highly sensitive to the PARP inhibitor olaparib. *Mol. Cancer Ther.* (2013) doi:10.1158/1535-7163.MCT-12-0950.
5. Korbel, J. O. & Campbell, P. J. Criteria for inference of chromothripsis in cancer genomes. *Cell* (2013) doi:10.1016/j.cell.2013.02.023.
6. Cortés-Ciriano, I. *et al.* Comprehensive analysis of chromothripsis in 2,658 human cancers using whole-genome sequencing. *Nat. Genet.* (2020) doi:10.1038/s41588-019-0576-7.
7. Scarpa, A. *et al.* Whole-genome landscape of pancreatic neuroendocrine tumours. *Nature* (2017) doi:10.1038/nature21063.
8. Alexandrov, L. B. *et al.* The repertoire of mutational signatures in human cancer. *Nature* (2020) doi:10.1038/s41586-020-1943-3.
9. Cameron, D. L. *et al.* GRIDSS, PURPLE, LINX: Unscrambling the tumor genome via integrated analysis of structural variation and copy number. *bioRxiv* (2019) doi:10.1101/781013.
10. Takizawa, N. *et al.* Molecular characteristics of colorectal neuroendocrine carcinoma; Similarities with adenocarcinoma rather than neuroendocrine tumor. *Hum. Pathol.* (2015) doi:10.1016/j.humpath.2015.08.006.
11. Konukiewitz, B. *et al.* Pancreatic neuroendocrine carcinomas reveal a closer relationship to ductal adenocarcinomas than to neuroendocrine tumors G3. *Hum. Pathol.* (2018) doi:10.1016/j.humpath.2018.03.018.
12. Vijayvergia, N. *et al.* Molecular profiling of neuroendocrine malignancies to identify prognostic and therapeutic markers: A Fox Chase Cancer Center Pilot Study. *Br. J. Cancer* (2016) doi:10.1038/bjc.2016.229.
13. Marabelle, A. *et al.* Association of tumour mutational burden with outcomes in patients with advanced solid tumours treated with pembrolizumab: prospective biomarker analysis of the multicohort, open-label, phase 2 KEYNOTE-158 study. *Lancet Oncol.* (2020) doi:10.1016/S1470-2045(20)30445-9.
14. Angus, L. *et al.* The genomic landscape of metastatic breast cancer highlights changes in mutation and signature frequencies. *Nat. Genet.* (2019) doi:10.1038/s41588-019-0507-7.
15. van Dessel, L. F. *et al.* The genomic landscape of metastatic castration-resistant prostate cancers reveals multiple distinct genotypes with potential clinical impact. *Nat. Commun.* **10**, 546051 (2019).

Summary of major revisions

- We re-performed all analysis and re-generated all figures based on the exclusion of a possible rare non-neuroendocrine occurrence within our cohort. This reduced our total sample size from $n = 86$ to $n = 85$.
 - o No major differences regarding the results and conclusions due to this exclusion were found following. All shown data, numbers, legends and figures are adjusted accordingly.
- We optimized our analysis for detecting mutually-exclusive genes, thereby revealing different genes within certain sub-populations such as *CSMD1* and *CSMD3* in the mNEC population. Due to the difference in selecting the initial genes to be investigated, *UBR5* and *ZFHX4* were no longer found enriched (previously found enriched within mNEC with adjusted $0.05 > p < 0.1$) as these did not satisfy the criteria of harboring either coding mutations *or* deep copy-number alterations in $\geq 20\%$ of the respective major subgroup. These were present in the previous analysis as they only contained aberrations in $\geq 20\%$ of the mNEC population when summarizing both coding mutations *and* deep copy-number alterations.
- We extended our investigation into significant differences of the observed mutational signatures between our major subgroups (**Suppl. Fig. 6g**) and report 5 mutational signatures with differing relative contribution.
- As part of the comments of the reviewer, we reshuffled parts of our manuscript to allow for easier and more concise reading. In particular, we now first introduce previous genomic-studies for NEN and their major finding (drivers) and put this into perspective with our results during the discussion.
- Due to reshuffling, as part of the reviewer's comments, we opted to combine our previous **Suppl. Fig. 1** and **Suppl. Fig 2** as parts of the previous **Suppl. Fig 1 (c-d)** have now been placed in **Fig. 1 (c-d)**.
 - o All references to figures have been corrected throughout the manuscript.

REVIEWERS' COMMENTS

Reviewer #1 (Remarks to the Author):

Regarding to trans-differentiation and genomic alterations of NEC, recently Kawasaki et al. published interesting results (An organoid biobank of rare human neuroendocrine neoplasms enables genotype-phenotype mapping. *Cell* 183(5):1420-35, 2020). They should cite this report and discuss how similar and different their results are from these data in terms of NEC and NEC differentiations.

Reviewer #2 (Remarks to the Author):

Many of the reviewers points of concern were addressed, one major concern remains and was not solved:

The mixture of many different entities, including neuroendocrine carcinomas of different organs as well as neuroendocrine tumors of different organs, leads to very small numbers of each tumor type in the end, limiting the room for significant novelties. Without matching primary tumors and metastases not much can be added on the understanding of metastases from a genomic standpoint. As a consequence, the authors report on the metastatic samples mainly genomic alterations that have been described before on primary tumors.

In my view, the major novelty is the lack of major differences in mNEN compared to primary tumors, this is not stated.

Specific:

Line 104-105: as now clearly indicated by the authors, a major novelty of the study is the sequencing of metastases, please indicate the number of metastases sequenced, the patients, where only the primary tumors are sequenced, do not add to this novelty.

Line 503: "reveals that the underlying genomic alterations could be exploited for better distinction of tumor subgroups..." by what? It is not clear to the reader what is the advantage of sequencing a metastasis compared to sequencing the primary tumor regarding detecting treatment options, which is one intervention less. This would be a major conclusion of the study. CSMD1 and CSMD3 could be involved in NEC metastasis, is there a treatment strategy available?

506-513: this statement is well true

Page 7, line 211: Association with smoking is very clear in lung NEC, however it is expected in analogy to non-neuroendocrine carcinomas to be present in bladder NEC, esophageal NEC.

A lack of pathological analysis of the metastases sequenced is lowering the value of associations to pathological data including grade. Assuming Ki-67 level as identical as in the primary tumor is likely leading to some errors, for PanNET it has been shown that 50% of synchronous as well as non-synchronous metastases show differences in Ki-67 levels.

Reviewer #3 (Remarks to the Author):

In this revised version the authors have greatly improved their manuscript mainly through a better organization of Introduction and Results. Accordingly, also Figures and cited literature have been refined.

There are only few remaining minor suggestions listed below:

Line 136: Is the quoted Figure 2 appropriate here?

Line 228: "... , we detect" should be replaced by "... , we detected".

Line 268: references should be quoted about the proper use of GISTIC2 and dN/dS , as mentioned in the reviewer's response.

Response to reviewers concerning: "The genomic landscape of 85 metastatic neuroendocrine neoplasms reveals subtype-heterogeneity and potential therapeutic targets" by van Riet et al.

Reviewer #1

Regarding to trans-differentiation and genomic alterations of NEC, recently Kawasaki et al. published interesting results (An organoid biobank of rare human neuroendocrine neoplasms enables genotype-phenotype mapping. *Cell* 183(5):1420-35, 2020).

They should cite this report and discuss how similar and different their results are from these data in terms of NEC and NEC differentiations.

Response:

We have indeed cited this very interesting study utilizing organoids derived from gastroenteropancreatic neuroendocrine neoplasms. We have compared the most striking (genomic) differences and chromosomal loss-of-heterozygosity between NEC and NET and overall found them comparative with frequencies from our cohort. E.g., both the GEP-NEC organoids and the CPCT-02 cohort confirm the enrichment of major drivers such as *RB1*, *TP53*, *APC* and *MYC* within NET tissue vs. NEC tissue. Whilst the GEP-NECs also occasionally harbor drivers enriched for other NEN-populations (e.g., *MEN1*). We have tried to relay that the GEP-NEN organoids depict a similar landscape as our cohort. (line 457 - 463)

Response to reviewers concerning: "The genomic landscape of 85 metastatic neuroendocrine neoplasms reveals subtype-heterogeneity and potential therapeutic targets" by van Riet et al.

Reviewer #2

Many of the reviewers' points of concern were addressed, one major concern remains and was not solved: The mixture of many different entities, including neuroendocrine carcinomas of different organs as well as neuroendocrine tumors of different organs, leads to very small numbers of each tumor type in the end, limiting the room for significant novelties. Without matching primary tumors and metastases not much can be added on the understanding of metastases from a genomic standpoint. As a consequence, the authors report on the metastatic samples mainly genomic alterations that have been described before on primary tumors.

In my view, the major novelty is the lack of major differences in mNEN compared to primary tumors, this is not stated.

Specific:

- 1) Line 104-105: as now clearly indicated by the authors, a major novelty of the study is the sequencing of metastases, please indicate the number of metastases sequenced, the patients, where only the primary tumors are sequenced, do not add to this novelty.
- 2) Line 503: "reveals that the underlying genomic alterations could be exploited for better distinction of tumor subgroups..." by what? It is not clear to the reader what is the advantage of sequencing a metastasis compared to sequencing the primary tumor regarding detecting treatment options, which is one intervention less. This would be a major conclusion of the study. CSMD1 and CSMD3 could be involved in NEC metastasis, is there a treatment strategy available?
- 3) 506-513: this statement is well true.
- 4) Page 7, line 211: Association with smoking is very clear in lung NEC, however it is expected in analogy to non-neuroendocrine carcinomas to be present in bladder NEC, esophageal NEC.
- 5) A lack of pathological analysis of the metastases sequenced is lowering the value of associations to pathological data including grade. Assuming Ki-67 level as identical as in the primary tumor is likely leading to some errors, for PanNET it has been shown that 50% of synchronous as well as non-synchronous metastases show differences in Ki-67 levels.

Response:

We fully agree that the incorporation of matched primary and (even multiple) metastatic tissues from the same patient would provide additional information on patient-specific somatic and treatment-induced evolution.

We also agree with the notion that no significant changes in the frequency of the major drivers can be found between primary and metastatic NEN; an observation which seemingly also holds true for additional malignant tissues.

We have highlighted this further within the discussion (line 508 - 509) and have also added, in our view, the benefit of whole-genome sequencing a metastatic biopsy compared to the primary malignancy.

Response - Specific comments:

- 1) We have added the exact number of metastatic ($n = 70$) vs. primary lesions ($n = 15$) (line 104 -105).

Response to reviewers concerning: "The genomic landscape of 85 metastatic neuroendocrine neoplasms reveals subtype-heterogeneity and potential therapeutic targets" by van Riet et al.

- 2) We have extended our reasoning of the importance of sequencing a metastatic lesion **(line 512 - 516)**.
- 3) The challenge of deciphering the non-coding somatic genome is currently being tackled within the scientific community and we hope that it is only a matter of time before we can delve further within the underlying non-coding mechanisms driving (m)NEN. This cohort will surely become of importance in this collaborative endeavor.
- 4) We do agree that the finding of a smoking signature could also indicate a primary neuroendocrine tumor location in the urinary bladder or esophagus, as well as in the lung. We have clarified in the manuscript that we at least do not have clues pointing to a non-neuroendocrine tumor of the lung such as a non-small cell lung cancer **(line 241 - 247)**.
- 5) We agree that Ki67 is a very heterogeneous marker, within and between tumor lesions. Therefore, the lack of analysis of Ki67 expression on all metastatic lesions can underestimate the true Ki67 expression of the sequenced metastatic tissue. Due to these difficulties, we specifically did not perform in-depth analysis on these data.

Response to reviewers concerning: "The genomic landscape of 85 metastatic neuroendocrine neoplasms reveals subtype-heterogeneity and potential therapeutic targets" by van Riet et al.

Reviewer #3

In this revised version the authors have greatly improved their manuscript mainly through a better organization of Introduction and Results. Accordingly, also Figures and cited literature have been refined.

There are only few remaining minor suggestions listed below:

- 1) Line 136: Is the quoted Figure 2 appropriate here?
- 2) Line 228: "..., we detect" should be replaced by "..., we detected".
- 3) Line 268: references should be quoted about the proper use of GISTIC2 and dN/dS, as mentioned in the reviewer's response.

Response - Specific comments:

- 1) We agree with the reviewer that "Suppl. Table 1" would be more appropriate in this context and have adjusted this accordingly (**line 135**).
- 2) We have corrected this misspelling (**line 266**).
- 3) We have indeed added these references in place. This was previously only done within the "Methods" section.